Geometric morphometric analysis of intratrackway variability: a case study on theropod and ornithopod dinosaur trackways from Münchehagen (Lower Cretaceous, Germany)

Lallensack Jens N. 1 jens.lallensack@uni-bonn.de
van Heteren Anneke H. 1 3
Wings Oliver 2
1 Division of Paleontology, Steinmann Institute, University of Bonn , Bonn , Germany
2 Niedersächsisches Landesmuseum , Hannover , Germany
3 Current affiliation: Sektion Mammalogie, Zoologische Staatssammlung München , München , Germany
Maschner Herbert
Electronic publication date: 2016 Jun 8
Publication date: 2016
Volume: 4
Electronic Location ID: e2059
Received 2016 Mar 4; Accepted 2016 Apr 29
Copyright: ©2016 Lallensack et al.
Copyright year: 2016
Copyright holder: Lallensack et al.
License: This is an open access article distributed under the terms of the Creative Commons Attribution License, which permits unrestricted use, distribution, reproduction and adaptation in any medium and for any purpose provided that it is properly attributed. For attribution, the original author(s), title, publication source (PeerJ) and either DOI or URL of the article must be cited.
License URL: https://creativecommons.org/licenses/by/4.0/

Keywords: Lower Cretaceous, Trackways, Germany, Dinosaur tracks, Geometric morphometrics, Theropods, Ornithopods, Fossil footprints, Photogrammetry

Funding: Europasaurus-Project 85 882 Oliver Wings is currently funded in the Europasaurus-Project (grant no. 85 882) by the Volkswagen Foundation in the grant initiative “Research in Museums.” The funders had no role in study design, data collection and analysis, decision to publish, or preparation of the manuscript.

==============================
A profound understanding of the influence of trackmaker anatomy, foot movements and substrate properties is crucial for any interpretation of fossil tracks. In this case study we analyze variability of footprint shape within one large theropod (T3), one medium-sized theropod (T2) and one ornithopod (I1) trackway from the Lower Cretaceous of Münchehagen (Lower Saxony, Germany) in order to determine the informativeness of individual features and measurements for ichnotaxonomy, trackmaker identification, and the discrimination between left and right footprints. Landmark analysis is employed based on interpretative outline drawings derived from photogrammetric data, allowing for the location of variability within the footprint and the assessment of covariation of separate footprint parts. Objective methods to define the margins of a footprint are tested and shown to be sufficiently accurate to reproduce the most important results. The lateral hypex and the heel are the most variable regions in the two theropod trackways. As indicated by principal component analysis, a posterior shift of the lateral hypex is correlated with an anterior shift of the margin of the heel. This pattern is less pronounced in the ornithopod trackway, indicating that variation patterns can differ in separate trackways. In all trackways, hypices vary independently from each other, suggesting that their relative position a questionable feature for ichnotaxonomic purposes. Most criteria commonly employed to differentiate between left and right footprints assigned to theropods are found to be reasonably reliable. The described ornithopod footprints are asymmetrical, again allowing for a left–right differentiation. Strikingly, 12 out of 19 measured footprints of the T2 trackway are stepped over the trackway midline, rendering the trackway pattern a misleading left–right criterion for this trackway. Traditional measurements were unable to differentiate between the theropod and the ornithopod trackways. Geometric morphometric analysis reveals potential for improvement of existing discriminant methods.

Introduction

The shape of a footprint is determined by the anatomy and the behavior (foot movements) of the trackmaker as well as the substrate properties (Falkingham, 2014). For many applications, especially for ichnotaxonomy and trackmaker identification, foot anatomy is the component of interest, and alterations of the footprint shape from the foot shape introduced by substrate and behavior are regarded as “extramorphological” noise (Peabody, 1948). The influence of extramorphological variability on footprint shapes is difficult to assess. The presence of anatomical details, most importantly digital pads, is probably the best available criterion for the assessment of footprint quality (Belvedere & Farlow, in press). However, even high-quality footprints can be greatly influenced by extramorphological, often behavioral variability, and significant deformation can occur even when fine anatomical details such as skin impressions are preserved (e.g. Currie, Nadon & Lockley, 1991, Fig. 4). Furthermore, it is unclear if a lack of phalangeal pads is necessarily related to poor footprint quality, as it in some cases might reflect the soft part anatomy of the trackmaker’s foot (Lockley, 1998; Lockley, Meyer & Moratalla, 1998). A thorough understanding of the influence of the anatomy, the behavior, and the substrate properties on footprint shape is therefore fundamental for any interpretation of fossil footprints, independent of their quality of preservation.

The influence of substrate and behavior becomes obvious within trackways. As foot anatomy can be considered constant within a trackway, at least amongst all footprints left by the same foot, in theory any variability can be attributed to behavior and substrate properties. In our case study, we analyze variability within three long and well preserved tridactyl trackways from the Lower Cretaceous Münchehagen locality in Germany, left by one ornithopod and two theropod trackmakers. We describe the morphology of the analyzed footprints, both qualitatively based on the best-defined footprint of the respective trackway, and quantitatively based on Procrustes mean-shapes. We then assess and review the variability, and thus informativeness, of track features commonly employed for ichnotaxonomy, trackmaker identification, and the discrimination between left and right footprints. Geometric morphometrics is utilized to measure variability and covariation of individual landmarks representing separate parts of the footprints. Principal component analysis reveals variation patterns that possibly are controlled by the trackmaker’s behavior. Criteria for the discrimination of left and right footprints are evaluated for both the theropod and the ornithopod trackways. We furthermore assess and discuss the ability of both linear measurements and geometric morphometric methods to discriminate between the ornithopod and theropod footprints. Although our analyses are based on interpretative, and thus subjective, outline drawings, we were able to reproduce important results using a fully objective approach to define footprint extents.

Figure 1 Map showing the location of Münchehagen in Germany.

History and Sedimentology of the Münchehagen Locality

The Wesling sandstone quarry in Münchehagen (municipality of Rehburg-Loccum, Lower Saxony, Germany) represents one of the most productive dinosaur track localities in Germany (Wings et al., 2012) (Fig. 1). The site belongs to the Obernkirchen Sandstone in the Lower Saxony Basin, which is Berriasian in age and traditionally attributed to the “German Wealden” (Erbacher et al., 2014; Mutterlose, Bodin & Fähnrich, 2014). According to the terminology recently introduced by Erbacher et al. (2014), the Obernkirchen Sandstone is part of the Barsinghausen Subformation within the Deister Formation, which falls within the Bückeberg Group (formerly known as Bückeberg Formation). In the Obernkirchen Sandstone, dinosaur tracks have been discovered since the nineteenth century (Grabbe, 1881; Hornung et al., 2012). The first track-bearing layer in Münchehagen was discovered in 1980 in an abandoned part of the Wesling quarry. Containing multiple trackways of sauropod dinosaurs, this site was granted national monument status, sheltered by a protective building, and included in the exhibition of the newly built Dinosaurier-Freilichtmuseum Münchehagen soon after its discovery (Hendricks, 1981; Fischer, 1998; Lockley, Wright & Thies, 2004; Wings et al., 2012). Multiple isolated slabs containing ornithopod pes and manus tracks have been recovered and described from overlaying strata (Lockley & Wright, 2001; Lockley, Wright & Thies, 2004; Gierlinski et al., 2008). Large-scale excavations of footprints from these overlaying strata are carried out since 2004 in the nearby active part of the Wesling quarry (Wings et al., 2012; Wings, Lallensack & Mallison, in press). In contrast to the old Wesling quarry, the new Wesling quarry prevailingly comprised tridactyl footprints referable to both ornithopod and theropod dinosaurs (Wings, Lallensack & Mallison, in press), although discoveries of sauropod tracks in the new quarry have been made recently.

In this case study, we analyze three trackways from the lower level of the new Wesling quarry, which have been previously assigned to theropod (trackways T2 and T3) and ornithopod (trackway I1) trackmakers (Wings et al., 2012; Wings, Lallensack & Mallison, in press). These trackways are amongst the longest and best preserved dinosaur trackways from Germany (Fig. 2). The T3 trackway was composed of 47 consecutive footprints. With a continuously recorded length of more than 55 m, it can be regarded as one of the longest dinosaur trackways in the world (cf. Xing et al., 2015). The I1 trackway, composed of 53 consecutive footprints, measured ca. 35 m in total length, while the 24 footprints of the T2 trackway account for a total length of 24.95 m. The trackways have been excavated between 2009 and 2011. Additional footprints of the T3 and possibly of the T2 trackway have been discovered in 2015, extending the length of these trackways even further. These new finds are not included in the present study.

Figure 2 Sitemap based on photogrammetric data showing the three analyzed trackways (T3, T2, and I1), which represent some of the longest dinosaur trackways from Germany.

The proximal sections of the T3 and I1 trackways, excavated before 2011, were not included because photogrammetric documentation is not available. Possible continuations of the T3 and T2 trackways discovered in 2015 are also not included.

The Münchehagen locality mainly exposes fine to medium quartz sandstones, which are brown to yellow-gray in color and strongly siliceously cemented. The dip of the beds is between 3° and 6° towards the west (Wings et al., 2012; Wings, Lallensack & Mallison, in press). Although laterally variable, the beds can be classified in 19 lithological units (LU) (Wings et al., 2012). Coaly layers covering parts of some of the bedding planes are attributed to plant detritus. Symmetrical ripple marks, generally orientated in north-south direction and classified as small-scale wave ripples, are present on most bedding planes, indicating that flow and wave direction were constant during deposition of several beds (Wings et al., 2012; Wings, Lallensack & Mallison, in press). Drainage structures on few bedding planes indicate a paleoflow direction to the west (Wings et al., 2012; Wings, Lallensack & Mallison, in press). The paleoenvironment has been interpreted as brackish with both freshwater and marine influences (Mutterlose, 1997).

The three trackways analyzed herein stem from LU7, which measures ca. 8 cm in thickness and encompasses fine to medium, very well sorted quartz sandstones topped by a thinly layered stack of silty mudstones (Wings et al., 2012). The true tracks are found on top of the silty mudstones, with undertracks preserved in the sandstones of LU7, LU6, and LU5, and natural track casts preserved on the underside of LU8 (Wings, Lallensack & Mallison, in press). The silty mudstones are very variable in thickness laterally, but do not exceed 3 cm. Within the footprints, the thickness of the silty mudstones accounts for less than a centimeter. The silty mudstones were destroyed during excavation in approximately 40% of the footprints due to their fragility, exposing the undertracks within the sandstones. Evidence for aerial exposure is lacking, raising the possibility that the layer was constantly covered by water during track formation.

Material & Methods

Data acquisition

The majority of the footprints analyzed herein was destroyed by weathering after their removal from the sediment layers, with only a section of the T2 trackway (T2/01–15) and several slabs with natural casts being preserved, including the excellently preserved casts of T2/4, T3/18 and T3/39–41 (Figs. 3 and 4). A section of the T2 trackway was re-assembled after excavation for display within the protective building of the Dinosaurier-Freilichtmuseum Münchehagen. The present study is mostly based on photogrammetric documentation carried out in 2011, which encompasses the analyzed sections of the T3 and I1 trackways, as well as part of the T2 trackway section. For the remainder of the T2 trackway, photogrammetric data collected in 2009 and a recent photogrammetry of the re-assembled and exhibited section of the trackway was used. Photogrammetric documentation was carried out by one of us (OW), using methods outlined in Mallison & Wings (2014). No photogrammetric data is available of the first sections of the T3 (T3/1–T3/23) and I1 (I1/1–I1/17) trackways, except for few excavated slabs that have been brought into the protective building of the Dinosaurier-Freilichtmuseum and are still accessible (see Wings et al., 2012, Fig. 13 for a sketch of the complete presumed trackway course). Five photogrammetric models of the trackway sections were created (T3/23–47; T2/01–15; T2/11–16; T2/16–24; I1/17–53) using the software Agisoft PhotoScan Professional 1.0.4 (www.agisoft.com). The horizontal planes were defined by setting three marker points on the surfaces of the models, respectively. To reduce the influence of the unevenness of the surfaces, the distance between the marker points was maximized. Individual footprints were cropped out of the trackway models for further analysis. Orthophotos of each trackway section were generated with Agisoft Photoscan. All surface meshes are reposited at Figshare (Table 1). The original photographs are available from OW on request.

Figure 3 Depth-color images of well preserved footprint casts of the T2 and T3 trackways.

(A) The cast of T2/4, which belongs to a left foot, is the best preserved cast of the T2 trackway. It shows an excellently preserved claw impression on digit III. (B) The cast of T3/18, belonging to a left foot, is the best preserved cast of the T3 trackway. It features well preserved claw impressions, phalangeal pads, and a complete heel region. (C) The casts of T3/39–T3/41. The respective moulds of T3/39 and T3/41 are illustrated in Fig. 5. (A) and (B) to scale. Color legend scales in mm.

Figure 4 Depth-color image of a slab containing the natural casts of I1/16 (lower left), I1/17 (upper left), and T3/22 (right).

The mould of I1/16 is illustrated in Fig. 7A. Color legend scale in mm.

Table 1 Photogrammetric surface models available via Figshare.

Footprints	Preservation	Status	URL	
T3/25–T3/48	Moulds	in situ, 2011	http://dx.doi.org/10.6084/m9.figshare.3027211	
I1/17–I1/53	Moulds	in situ, 2011	http://dx.doi.org/10.6084/m9.figshare.2972329	
T2/1–T2/15	Moulds	Excavated and reassembled	http://dx.doi.org/10.6084/m9.figshare.3025144	
T2/11–T2/16	Moulds	in situ, 2009	http://dx.doi.org/10.6084/m9.figshare.3026863	
T2/16–T2/24	Moulds	in situ, 2011	http://dx.doi.org/10.6084/m9.figshare.3027067	
T3/39–T3/41	Natural casts	Excavated and reassembled	http://dx.doi.org/10.6084/m9.figshare.3027949	
T3/22; I1/16–I1/17	Natural casts	Excavated	http://dx.doi.org/10.6084/m9.figshare.3027553	
T3/18	Natural cast	Excavated	http://dx.doi.org/10.6084/m9.figshare.3027385	
T2/4	Natural cast	Excavated	http://dx.doi.org/10.6084/m9.figshare.3029698	

For the morphometric analysis, we selected 13 well preserved footprints of the T3 trackway (Fig. 5), 8 of the T2 trackway (Fig. 6), and 17 of the I1 trackway (Fig. 7). Criteria for the selection included the presence of a largely intact layer of silty mudstones, therefore excluding undertracks. Furthermore, only footprints were selected that could be fully defined by a single contour line; this excludes any shallow and incomplete footprints. Contoured depth-color images were created for each selected individual footprint with the open source software Paraview 4.1 (www.paraview.org). For each image, a fixed number of 30 contour lines are used, whose spacing is relative to the total height of the cropped-out footprint model along the z-axis. Therefore, the same contour lines will represent different total depths in separate images. This approach of relative contour lines allows for the direct comparison of individual contour lines in separate models, independent of the depth and size of the footprints. All left footprints were mirrored to fit the right ones. Outlines were traced on the contoured depth-color images using the open source software Inkscape (www.inkscape.org).

Figure 5 Contoured depth-color images with interpretative outlines of the 13 footprints of the T3 trackway analyzed herein.

All footprints are to scale, and all left footprints were mirrored to fit the right ones. The color scale ranges from the lowest point (blue) to the highest point of the model (red); given the different depths in separate models, the depth of a specific color varies. (A) T3/23. (B) T3/26. (C) T3/29. (D) T3/34. (E) T3/35. (F) T3/36. (G) T3/37. (H) T3/39. (I) T3/40. (J) T3/43. (K) T3/44. (L) T3/46. (M) T3/47. Color legend scales in mm.

A site map (Fig. 2, Fig. S3) was drawn directly from the combined orthophotos and elevation models of all three trackways. A preliminary sitemap has already been published by Wings et al. (2012, in press, Fig. 13). This map appears to be deformed, and compared to our map is stretched to a degree of ca. 12%. The original numbering scheme published in Wings et al. (2012) is inconsistent in the T3 and I1 trackways, a result of the complex excavation process between 2009 and 2011. We therefore felt the need to develop a new consistent numbering scheme, which is already employed in Wings, Lallensack & Mallison (in press) for the T3 trackway. The new numbers are always notated in the form trackway/number, to avoid any confusion with the old scheme.

Figure 6 Contoured depth-color images with interpretative outlines of the eight footprints of the T2 trackway analyzed herein.

All footprints are to scale, and all left footprints were mirrored to fit the right ones. The color scale ranges from the lowest point (blue) to the highest point of the model (red); given the different depths in separate models, the depth of a specific color varies. (A) T2/01. (B) T2/11. (C) T2/12. (D) T2/13. (E) T2/14. (F) T2/21. (G) T2/22. (H) T2/24. Color legend scales in mm.

Definition of footprint margins: Interpretative and objective methods

The determination of the margin of a footprint can involve a problematical degree of subjective interpretation, compromising any quantitative analysis of footprint shape (Falkingham, in press). This problem most drastically affects footprints that lack well-defined phalangeal pads, fade gradually into the surrounding sediment and/or show multiple edges (Sarjeant, 1975; Thulborn, 1990; Falkingham, 2010), which is the case in many deeper dinosaur tracks. Subjectivity equally affects both measurements and outline drawings, and both the size and shape of a footprint can differ considerably when separate approaches are employed. Previously adopted approaches range from tracing along the margin between the flat floor and the sloping wall of the footprint (minimum outline approach, Martin et al., 2012) to tracing at the level of the surface of the surrounding sediment (Falkingham, 2010). Only few studies elaborate on the criteria used to define the outlines (for exceptions, see e.g., Pittman & Gillette, 1989; Belvedere, 2008; Martin et al., 2012), making comparisons of published outline drawings and measurements ambiguous. Although a number of objective approaches for the definition of footprint margins exist (Falkingham, in press), all have practical problems and so far are rarely employed in actual studies.

All interpretational outlines were performed by one of us (JNL) in order to avoid interobserver variability. In order to reduce subjectivity in our interpretational outlines, we followed three criteria, namely the steepest slope, a consistent elevation, and the maximization of digit length. To achieve the latter, the most proximal slope of the sediment bar separating the digit impressions is interpreted as the hypex point when multiple slopes are present. As all three criteria cannot be completely fulfilled at the same time, a “best guess” approach was adopted, attempting a best fit between all three criteria. Exceptions were made to exclude extramorphological features (T3/44; Fig. 5K) and to include partly filled digit impressions (I1/32; Fig. 7D). Despite these criteria, many decisions made in drawing the outlines were rather ambiguous. To check for an interpretational bias of our results, we defined two alternative, objective approaches to determine landmark positions, as described below.

Figure 7 Contoured depth-color images with interpretative outlines of the 17 footprints of the I1 trackway analyzed herein.

All footprints are to scale, and all left footprints were mirrored to fit the right ones. (A) I1/17. (B) I1/30. (C) I1/31. (D) I1/32. (E) I1/35. (F) I1/36. (G) I1/38. (H) I1/39. (I) I1/40. (J) I1/41. (K) I1/45. (L) I1/46. (M) I1/48. (N) I1/49. (O) I1/50. (P) I1/52. (Q) I1/53. Color legend scales in mm.

Our first objective approach is based on the steepest slope (the turning point of the surface inclination according to Ishigaki & Fujisaki (1989), equivalent to the inflection point) of the footprint wall, henceforth called the steepest slope approach. The steepest slope is probably the most frequently used criterion for the definition of the margin of a track, although its determination by the naked eye of the observer usually is subjective (Ishigaki & Fujisaki, 1989). Implementing this criterion, we consequently selected the middle one of the three most closely spaced contour lines along the digital and hypex axes, respectively. To avoid that ripple marks of the surrounding sediment are traced rather then the footprints, only the lower two thirds of the 30 contour lines were used. Our second objective approach employs a specific contour line, and henceforth is dubbed the contour line approach. Since various methods for the three-dimensional digitalization of footprints have become available to ichnologists, contour lines have been repeatedly used instead of traditional interpretational drawings in order to reduce subjectivity (e.g., Petti et al., 2008; Falkingham, 2010; Romilio & Salisbury, 2014). In our approach, we placed landmarks on the intersection of the contour line with the digital and hypex axes in all of the tracks. Only the 11th and 12th contour lines were able to capture the complete outline in all T3 footprints selected for geometric morphometric analysis; the analysis was executed for both. Both objective approaches were performed on the T3 footprints.

An interpretational outline drawing usually aims to reproduce the anatomy of the trackmaker’s foot as best as possible, while minimizing the influence of intratrackway variability. The objective approach showing the lower overall variability of the landmark positions is therefore less influenced by intratrackway variability. This more repeatable approach was then equally performed on the T2 and I1 trackways (Figs. S1 and S2), and the results compared with those based on the interpretational outlines.

Trackway parameters and traditional morphometrics

Trackway parameters (i.e., the pace and stride lengths, pace angulation, and footprint rotation) were calculated from xy-coordinates taken from photogrammetric data by employing trigonometric functions. The measurement of trackway parameters requires the selection of a homologous reference point on the footprints. To analyze the influence of reference point choice on the measured values, we measured all trackway parameters twice, once with the tip of digit impression III and once with the base of digit impression III as reference point. In practice, the reference point on the base of digit impression III is difficult to measure in the more poorly preserved footprints (Thulborn, 1990). To obviate this problem, we measured the free length of digit impression III in a well-preserved footprint of the given trackway. Then we determined the position of the reference point in all footprints of that trackway according to the distance measured in the well preserved footprint from the tip of digit impression III down the footprint long axis (Fig. 8). Unless noted otherwise, all trackway parameters and speed estimates given below are based on coordinates determined by this procedure.

Footprint mean shapes of all three trackways were measured according to Moratalla, Sanz & Jimenez (1988), in order to determine the ability of traditional measurements to discriminate between theropod and ornithopod footprints (Fig. 8). Hip height of the trackmakers was assumed to equal 4 times footprint length, following Alexander (1976). Despite its simplicity, this relationship was shown to be the most accurate of various approaches (Henderson, 2003). The approximate body length of the theropod trackmakers was calculated using the average ratio of hip height to body length proposed by Xing et al. (2009), which is 1:2.63. For speed estimates, we employed the formula of Alexander (1976) for trackways with a S∕h (relative stride length) of <2.0: v≈0.25g0.5S1.67h−1.17

where v = locomotion speed [m∕s], g = gravitational constant (9.81 m∕s2), S = stride length [m], and h = hip height [m].

Geometric morphometrics

For the geometric morphometric analysis, six landmarks were defined (Fig. 8A). Landmarks 1, 3, and 5 are located at the tips of the digit impressions (i.e., the ends of the digital axes of digits II, III and IV, respectively). Landmarks 2 and 4 represent the hypex positions (i.e., the midpoints between digit impressions II and III, and III and IV). Landmark 6 is located on the heel and defined as the intersection of the footprint long axis with the proximal heel margin. The footprint long axis corresponds with the axis of digit impression III, following Leonardi (1987). Given the straight morphology of digit III in the examined footprints, the mediolateral position of landmark 6 also roughly corresponds to the midpoint of the footprint when measured between digit impressions II and IV. All landmark placements were performed by one of us (JNL), in order to avoid interobserver variability.

The six landmarks were digitized for each outline using the freeware tpsDig 2.17 (Rohlf, 2014). In order to calculate detailed mean shapes, six curves of equally spaced semi-landmarks were placed along the outline using tpsDig, each connecting two adjacent landmarks. To allow for an equal distribution of the semi-landmarks across all six curves, the number of semi-landmarks was calculated for each curve by measuring the relative lengths of each curve for each outline. These measurements were done using the “Measure Path” function in Inkscape. In total, 114 semi-landmarks were used for the T2 and T3 and 101 for the I1 outlines. The curves were converted to regular landmarks using tpsUtil 1.85 (Rohlf, 2014) and subjected to Generalized Procrustes Analysis (GPA) in the open source software MorphoJ 1.06c (Klingenberg, 2011). The GPA is a statistical analysis that provides a best fit between the footprint shapes by translating, rotating and scaling, eliminating any information but the mere shape of the footprints. GPA was shown to be the most accurate of several approaches for the estimation of mean shapes (Rohlf, 2003). We did not slide the semi-landmarks as part of the GPA, as we regard the curve as the unit of homology, not the individual semi-landmark. Vector plots were created using tpsRelw 1.54 (Rohlf, 2014), in order to illustrate the variability along the outline of the Procrustes mean shape, the average shape of all analyzed footprints. The vector arrows originating from each landmark and semi-landmark of the mean shape illustrate the deviations of each of the individual outlines.

It can be argued that the scaling step of the GPA is undesirable when calculating footprint mean shapes for single trackways, as all footprints stem from the same individual. Testing the influence of scaling, the digitized landmarks and semi-landmarks of the T3 footprints were imported into the open source software PAST 3.01 (Hammer, Harper & Ryan, 2001), which provides an option for rescaling the individual outlines to their original sizes after performing the GPA. The coordinates of the resulting mean shape then were imported into MorphoJ and compared with the standard GPA mean shape. The Procrustes distance between both shapes is negligibly small, accounting for 3.62⋅10−6. For comparison, the Procrustes distance between the T3 and I1 mean shape is as high as 6.1⋅10−3. GPA, therefore, can be used instead of a partial Procrustes analysis (which retains the original size), even though the deviation can be expected to be greater in small sample sizes and with very large variability in footprint shapes.

Principal component analyses (PCA) were carried out using MorphoJ to assess co-variation of separate footprint regions. Only the six landmarks were used, not the semi-landmarks, in order to reduce noise. Separate analyses were performed for each trackway, revealing trackway-specific variation patterns and allowing for the quantification of the variability of individual landmarks. The calculated Procrustes mean shapes were imported into MorphoJ in order to make use of the warped outline drawing function, allowing for a better visualization of shape deformations. An additional PCA incorporates all three trackways in order to investigate whether PCA is capable of separating these trackways. For all PCAs, only the first three principal components were taken into account as they describe the majority of the total variation. Furthermore, a canonical variate analysis (CVA) was carried out using MorphoJ, which again was restricted to the six true landmarks.

Results

Objective methods for the definition of footprint margins

When based on interpretational outlines, the Procrustes-fitted landmarks of the selected T3 footprints show a variance of 0.027. Employing the objective contour line approach as described above, the total variance is markedly higher, accounting for 0.0453 for the 12th and 0.0457 for the 11th contour line. This higher variance suggests an increased influence of substrate properties and behavior on the individual outlines. The steepest slope approach resulted in a total variance of 0.0379, and therefore falls between the interpretational and the contour line approach. Of both objective approaches, the steepest slope approach is therefore least affected by intratrackway variability.

Results from the objective steepest slope approach for all three trackways were then compared with those of the interpretational approach. For the two theropod trackways, landmark positions of both approaches are reasonably close to each other in most cases, with few outliers. Consequently, the recorded shape changes are very similar, with notable differences occurring only in PC2. In PC1, both approaches show an anterior shift in the position of the heel together with a posterior shift in the position of the lateral hypex, while other landmarks are rather stationary (Fig. 12, Fig. S2). In the steepest slope approach, PC1 describes 53% in the T3 and 66% in the T2 trackway, which is higher than in the interpretative approach (46% and 55%, respectively; Fig. 11, Fig. S1).

For the I1 footprints, landmark positions of both approaches differ markedly in many examples (Fig. S2). Particularly large differences occur in the position of the hypices; in many I1 footprints, both a proximal and a distal slope, often separated by an extensive plateau, can qualify as a landmark position. Consequently, the resulting mean shape of the steepest slope approach is less well defined, with the hypex positions located more distally, significantly reducing the free length of the digit impressions. In two footprints (I1/49, I1/32), partially filled digit impressions caused the steepest slope approach to drastically underestimate digit lengths. Despite these differences in landmark placement, PC1 and PC2 of both the steepest slope and the interpretative approach concur that the lateral hypex and the heel do not show an increased variability, in contrast to the theropod footprints. However, PC1 of the steepest slope approach shows a pronounced proximal displacement of the position of the medial hypex, in contrast to the interpretative approach. Again, the eigenvalue of PC1 is higher in the steepest slope approach, amounting to 33%, compared to 28% in the interpretative approach. PC2 describes 30% of the total variability in the steepest slope approach; as in the interpretative approach, this value is only slightly smaller than that of PC1.

Figure 8 Landmarks and measurements of footprint shapes used in this study, exemplified on the T3 trackway.

(A) Footprint sketch indicating landmark placement. Landmarks 1, 3, and 5 represent the tips of the digit impressions and are placed at the endpoints of the digital axes of digit impressions IV, III, and II, respectively. Landmarks 2 and 4 represent the hypex positions, and are placed at the midpoints between digit impressions III and IV and II and III, respectively. Landmark 6 captures the heel, and is defined as the intersection of an extension of the digital axes of digit impression III with the outline. (B) The measuring scheme proposed by Moratalla, Sanz and Jimenez (1988), applied to the T3 mean shape, adapted from Moratalla, Sanz and Jimenez (1988) and Romilio and Salisbury (2011). Measurements are done in order to determine the ability of traditional measurements to differentiate between the theropod and ornithopod footprints analyzed herein. Deviations from the original scheme are discussed in the text. Abbreviations: L, footprint length; W, footprint width; K, M, heel-interdigital distances; LII–IV, digit lengths; BL2–4, basal digit lengths (digit free lengths); WBII–IV, basal digit widths; WMIII-IV, middle digit widths. (C) Measured trackway parameters. Reference points for the pace- (LP, RP) and stride (S) lengths are determined by measuring a fixed distance down the axis of digit impression III (DAIII), reducing the influence of variation in footprint rotation, as discussed in the text. The trackway midline (TML) is parallel to the opposing stride of the footprint in question. Pace angulation (γ) is the angle between the left and right paces, and footprint rotation is the angle between the axis of digit impression III and the opposing stride.

Qualitative and quantitative description of footprint shapes

Trackway T3. Traditionally, footprint morphology is often assessed by describing and illustrating the best preserved or most representative footprints of a sample (Thulborn, 1990). In the T3 trackway, footprint morphology is best seen in the well preserved natural cast of T3/18 (Fig. 3). With a length-to-width ratio of 1.08, this footprint is only slightly longer than wide even when the weakly impressed metatarsophalangeal pad of digit IV is taken into account. The interdigital angle between digit impressions III and IV (43.5°) is larger than that between digit impressions II and III (33.7°). Digit IV is impressed along its whole length, although only the part proximal to the metatarsophalangeal pad is deeply impressed, with the metatarsophalangeal pad appearing very shallow. The deeply impressed part is similar in proximal extension to that of digit impression II; digit impression III is much shorter proximally. Three well defined phalangeal pads are present in digit impression IV. On digit II and III, the phalangeal pads are deformed obliquely, and are not readily distinguishable. Digit impression IV is more narrow than the other two digit impressions, which are similar in width. Digit impression III is the deepest, particularly in its distal part; digit impressions III and IV are connected with each other at their bases, while digit impression II appears isolated. Distinct claw marks are present in all three digit impressions, but vary greatly in size and position. The claw mark of digit impression II is the largest, being located lateral to the digital axis and slightly curved medially. In digit impression III, the claw mark appears much shorter, is located medial to the digital axis and curved towards the medial side. The claw mark of digit impression IV is less well pronounced, being located lateral to the digit axis. Claw marks of digits II and IV are thus strongly asymmetric in their position. The heel region is V-shaped, with a slight medial indentation below the base of digit impression II.

We subjected 13 selected natural molds to a geometric morphometric analysis. The resulting mean shape, although based on footprints of a generally lesser quality than the natural cast of T3/18, is a very informative quantification of footprint shape, as intratrackway variability is leveled out. In the molds, digital pad impressions are frequently preserved, but only in few footprints a nearly complete set is discernible (best seen in T3/26, Fig. 5B). In most footprints the pad impressions are interior topographical features that do not contribute to the traced outline, and thus do not contribute to the mean shape. This often leads to a steplike topography, where a maximization in the steepness of the slope delimits the footprint extents, and a second maximization in the steepness delimits the digital pads. The position and orientation of the deeply impressed bottom of the digit impressions does not always correspond to the traced footprint margins (Figs. 5A, 5C, 5D and 5I). In footprint T3/34 (Fig. 5D), the deeply impressed bottom of digit impression III even appears to follow an arc, while the traced outline of this digit is relatively straight. This might be attributed to foot movements, indicating that the latter have a great influence on the shape of the footprint.

The T3 mean shape is wider than long (length-width ratio: 0.92), a consequence of the incomplete or lacking impression of the metatarsophalangeal pad of digit impression IV in most of the footprints. In comparison with the natural cast of T3/18, the heel is much broader and more asymmetrically shaped, with the lateral bulge protruding beyond the medial bulge. The mean shape is dominated by the robust digit impression III, which is significantly longer and wider than the impressions of digits II and IV. The free length of digit impression III accounts for 61% of the total length of the shape. The digit impressions are straight, and indentations indicating phalangeal pads are not clearly visible. The apex of digit III, marking the position of the claw mark, is located medially to the digital axis, giving the digit impression an asymmetrical shape, with the lateral margin of the distal half forming a broad arc. The distal end of digit II appears to be symmetrical, while the distal end of digit IV shows an apex laterally to the digital axis, inverting the condition seen in digit III. The lateral hypex is located posteriorly relative to the medial hypex; accordingly, the free length of digit IV is greater than that of digit II. The depth of the selected footprints varies from 38 to 72 mm, with a mean of 51 mm and a coefficient of variation of 0.17. A significant correlation between maximum footprint depth and footprint shape could not be observed, which might be due to the limited sample size.

Trackway T2. The well preserved natural cast of T2/22 may be regarded as one of the best preserved ichnites known from the T2 trackway. This footprint shows a length-width ratio of 1.23. Digital pads are indistinct in most footprints, although T2/22 probably shows two pads in digit impression II and four pads in digit impression IV, including the metatarsophalangeal pad. Digit impression III of the mean shape shows a slight constriction at around half of its free length, possibly delimiting the two distalmost phalangeal pads of this digit. The T2 mean shape, which is based on eight selected footprints, is only slightly longer than wide (length-width ratio: 1.04), due to the incomplete impression of the heel in most footprints. It is more gracile in appearance compared to the T3 mean shape, mostly due to the slender and elongated digit impression III. Digit widths are similar in all three digits, unlike the T3 mean shape where digit III is wider. Digit III is substantially longer than digits II and IV; its free length accounts for 64% of the total footprint length, similar to the T3 mean shape. Digit II is longer than digit IV and shows a slight medial bend at midlength, while the distal end is symmetrical. Digit III is straight. Claw marks of digit III are either located centrally (e.g., T2/1) or medial with respect to the digital axis (e.g., T2/4, Fig. 3). In the mean shape, the medial tendency of the claw mark is recorded by the asymmetrical morphology of the tip of digit III, although this asymmetry is less pronounced than in the T3 mean shape. Digit IV is straight, with the apex of the distal end being located laterally to the digital axis, suggesting a lateral position of the claw mark. This lateral position is best seen in the well preserved cast of T2/4 (Fig. 3). The hypices do not show any offset, unlike the T3 mean shape. The maximum depth of the selected footprints varies between 29 and 59 mm, with a mean of 36 mm and a coefficient of variation of 0.27. Thus, the depth is more variable than in the T3 trackway. Digit III usually represents the most deeply impressed part of the footprints.

Trackway I1. While the overall footprint proportions remain similar within the I1 trackway, shape varies considerably. A representative footprint is difficult to designate, but I1/36 (Fig. 7F) might show the best approximation of the trackmaker’s anatomy, given the degree of detail shown by this footprint. The following description is based on both this footprint and the mean shape. The length-width ratio of the mean shape is 0.91. The heel, rather than digit impression III, is the most prominent feature of the shape. Digit impression III is the longest digit, but the difference in length between digit III and digits II and IV is not as great as in the theropod footprints. Its free length accounts for only 48% of the total footprint length, less than in the theropod footprints. Digit impressions II and IV are both bended anteriorly, resulting in a crescent-like shape. At their bases, digit impressions II and IV protrude nearly horizontally from the heel region in the mean shape. Their posterior margin shows a pronounced kink at about midlength, resulting in a more anteriorly directed distal half of the digits. The bend in the anterior margin is less pronounced. The apex of digit impressions II and IV is located laterally and medially to the digital axis, respectively. Thus, the posterior margins of the distal thirds of both digit impressions are rounded, while the anterior margins are more straight. Digit impression III is straight and slightly tilted laterally; its apex tapers to a blunt ungual mark. The anteroposterior position of the lateral and medial hypex is equal. The heel shows a broad semicircular extension, which is separated from the bases of digits II and IV by an embayment on either side of the footprint. The semicircular extension is shifted laterally relative to the remainder of the shape. The footprints are similar in depth to the T3 footprints, varying from 41 to 77 mm in depth, with a mean depth of 55 mm and a coefficient of variation of 0.19.

Comparison of mean shapes. Two additional GPAs were performed to compare the T3 mean shape with the T2 and I1 mean shapes (Fig. 9C). The strongest shape difference between the T2 and the T3 mean shape occurs in digit impression III. In the T3 mean shape, this impression is both shorter and broader, with the main difference in width occurring on the medial margin. Thus, digit impression III is located more closely to digit impression II than to IV in the T3 shape, while the reverse is true for the T2 shape. This medial thickening of digit impression III in the T3 shape seems to be directly associated with a more anterior position of the medial hypex. The asymmetry of the termination of digit impression III is more pronounced in the T3 mean shape. Digit impression II is shorter in the T3 shape, while digit impression IV is laterally expanded on its distal end. The heel is best defined in the T2 shape and is more leveled in the T3 shape. The greatest difference between the T3 and I1 shapes (Fig. 9B) can be seen in the heel region. In the T3 shape, the heel shows two distal pads separated by a central embayment, with the lateralmost pad being located more proximally than the medial one. In the I1 shape, one single large heel pad is present, which on both sides is separated from the digit impressions by an embayment. Furthermore, the interdigital angle between digit impressions II and III is smaller in the T3 shape than in the I1 shape. The distal end of digit III is asymmetrical in the T3 shape, but symmetrical in the I1 shape, and the lateral hypex is placed more posteriorly in the T3 shape.

Figure 9 Discrimination between the footprint shapes of the T3, T2, and I1 trackway.

(A) Principal component analysis (PC1 vs. PC2) of the T3 (red), T2 (blue) and I1 (green) shapes based on six landmarks. 90% confidence ellipses are shown for each point cloud. High loadings on the first principal component indicate a more posterior position of the lateral hypex and a more anterior position of the heel landmark, while high loadings on the second principal component indicate a more slender footprint and a more posteriorly located medial hypex. (B) Canonical variate analysis (CV1 vs. CV2) of the T3, T2, and I1 shapes based on six landmarks. The best separation between the ornithopod and the theropod trackways is reached by CV1, high values of which describe a more posterior placement of the hypices, a more anterior position in the heel, and an anteriorly extended digit impression III. Coloring and confidence ellipses as in (A). (C) Procrustes-fitted mean shapes, allowing for pairwise comparisons (T3 vs. I1, T3 vs. T2, and T2 vs. I1). The T3 shape is shown in a continuous red line, the T2 shape in a blue dashed line with alternating short and long segments, and the I1 trackway in a dashed green line. Note that the shapes, while visually distinct, are very similar in their proportions. (D) Procrustes fitted pairs of mean shapes (continuous red line) and their respective mirror images (dashed blue line), highlighting footprint asymmetry. Note the pronounced asymmetry in the I1 ornithopod mean shape.

Variability of footprint shape

Variability in the heel. The heel region is highly variable in the T3 footprints in both extension and morphology. In the well preserved natural cast T3/18, the heel is fully impressed, showing a V-shaped morphology with an rounded proximal apex representing the metatarsophalangeal pad of digit IV (Farlow et al., 2000). This pad shows a slight lateral displacement relative to the axes of digit impression III; the medial side of this footprint features a weakly pronounced indentation directly below the proximalmost phalangeal pad of digit II. The impression of the metatarsophalangeal pad is significantly shallower than the phalangeal pads. Several of the natural molds show a similar morphology (T3/26, T3/40, T3/43–44, T3/46–48). In these footprints, the metatarsophalangeal pad can be located more centrally (T3/26, T3/43) or more laterally (T3/40; T3/47) with respect to digit impression III. In T3/47, the proximal part of digit II is not impressed, resulting in a very pronounced medial indentation. In the remaining footprints, the metatarsophalangeal pad of digit IV is not or only partly impressed, resulting in a foreshortened footprint with a broader rather than V-shaped heel. In some examples (T3/23, T3/39), both digits II and IV are shortened proximally to an equal degree, maintaining the asymmetry typical for theropod footprints. In other examples (T3/29, T3/36), digit II is fully impressed and more extensive than digit IV, which is shortened proximally, reversing the asymmetry seen in most of the other footprints. Finally, the shortest footprints result from incomplete impressions of both digits II and IV (T3/35, T3/37, T3/45). Most frequently, this results in a sub-symmetrical, bilobed morphology with a central indentation below digit impression III. These variations in the extent and shape of the heel strongly influence any associated measurements; for example, the coefficient of variation for footprint length is as high as 0.1 in the 13 analyzed T3 footprints, while it is only 0.04 for footprint width in the same footprints.

The heel is poorly impressed in most T2 footprints. In well preserved examples with pronounced phalangeal pads, the metatarsophalangeal pad is slightly displaced laterally relative to digit impression III, while digit impression II is distinctly shorter than digit impression IV, resulting in a very pronounced, step-like medial concavity. In the majority of footprints, the metatarsophalangeal pad is not or only partly impressed, shortening the impression of digit IV. When this pad is absent, proximal extension is similar in digit impressions II and IV; in this case, the medial concavity has transformed into a central concavity below digit III, giving the rear margin of the footprint a bilobed, subsymmetrical appearance. Few footprints show an extended, U-shaped heel impression, without any indication of a medial concavity.

Landmark analysis indicates that the extension of the heel is relatively stable in the I1 footprints. However, position and morphology can vary considerably. Typically, the heel pad forms a subcircular impression separated from the digital pads by a lateral and medial indentation, as shown by the I1 mean shape. In the individual footprints, the heel can appear V-shaped (I1/17, I1/53) or broadly rounded (I1/45, I1/50, I1/32, lacking the medial and lateral indentations. At the other extreme, the heel pad can be sub-rectangular in shape, with its lateral and medial sides forming angles of almost 90° to the impressions of digits IV and II, respectively (e.g., I1/28, I1/30). The position of the heel pad varies from being located centrally I1/31, I1/48) to strongly laterally (e.g., I1/30, I1/49, I1/53) with respect to digit impression III.

Variability in the digit impressions. In the T3 trackway, digit impressions are generally straight. An exception can be seen in T3/47 (Fig. 5M), where digit impression IV appears to be bowed laterally. Digit impression III is frequently widened; in T3/36, the absolute width of digit III exceeds that of the preceding footprint (T3/35) of more than 50% (Figs. 5E and 5F). Claw positions vary greatly. The well preserved natural cast of T3/18 shows a massive digit impression II with a subrectangular termination featuring a large claw mark located laterally with respect to the digital axis. This morphology is seen in several additional some footprints (best seen in T3/39). In other footprints (e.g., T3/39, T3/40, T3/26; Figs. 3 and 5), the claw mark can be located centrally, creating V-shaped digit terminations reminiscent of those recently described for the ichnogenus Bellatoripes (McCrea et al., 2014). In yet other footprints, the claw mark of digit II is located medially to the digital axis (best seen in T3/37, Fig. 5G). In digit impression III, the claw mark usually is located medial to the digital axis, but frequently is located centrally. In digit impression IV, the mean shape indicates a preferred claw mark location lateral to the digital axis.

In contrast to the T3 footprints, several of the digit impressions in the T2 trackway are very narrow and irregular slit-like, most strikingly digit III in T2/21 (Fig. 6F). Variability and position of the claw marks is similar to that observed in the T3 trackway. The T2 mean shape mirrors the condition seen in the T3 mean shape, with a medial tendency in the tip of digit impression III and a lateral tendency in the tip of digit impression IV, resulting either from the foot kinematics or preferred claw orientations. Exceptionally preserved claw impressions can be seen in digit impression III of the natural cast of T2/4 (Fig. 3A), where it is located medial to the digital axis, and T2/1 (Fig. 6A), where it is located centrally.

Variability in digit impression morphology and dimensions is particularly striking in the I1 trackway. The width of digit impression III at mid-length ranges from 14% (I1/38; Fig. 7G) up to 33% (I1/17; Fig. 7A) of footprint width. Digit impressions II and IV are generally rather narrow, with an abrupt bend at around midlength, as shown by the mean shape and the well preserved footprint I1/36 (Fig. 7F). In other footprints of this trackway, these digit impressions can be narrow and straight, lacking the bend (best seen in I1/45, Fig. 7K). These impressions can also appear short and thick, approaching a cloverleaf-shape (e.g., I1/17 and 30, Figs. 7A and 7B). The outer margin of digit impressions II and IV can display two pronounced steps, the first resulting from the subrectangular heel impression and the second from the bend at the midlength of the digit impression (e.g., I1/30, 31 and 36; Figs. 7B, 7C and 7F). In other footprints, the same margin can appear as a straight line (e.g., I1/17, 45 and 53; Figs. 7A, 7K and 7Q).

Figure 10 Procrustes-fitted coordinates of the analyzed trackways.

On the left-hand side, the mean shape is indicated by thick blue dots and the individual outlines as small black dots. On the right hand side, the variability relative to the mean shape is indicated by vector arrows. (A) Trackway T3. (B) Trackway T2. (C) Trackway I1.

Quantification of footprint shape variability. Landmark analysis was carried out on the selected footprints of all three trackways. Vector plots are used to illustrate the deviations of the Procrustes-fitted landmark and semi-landmark coordinates of the individual footprints from the respective coordinates of the mean shape. For the T3 footprints, the vector plot indicates a high variability in the lateral hypex, in the entire posterior margin of the heel region, and in the medial hypex (Fig. 10A). Digit shape, on the other hand, appears to be comparatively stable. The vector plot of the T2 shape reveals a high variability in the lateral hypex and in the right half of the heel region (Fig. 10B). Digit width also appears to be very variable. The I1 vector plot, on the other hand, indicates that the variance is more equally distributed along the outline, lacking variation hotspots in the hypex and heel as seen in the theropod footprints (Fig. 10C).

Figure 11 Loadings of the principal components of the T3, T2 and I1 trackways.

For the two theropod (T3 and T2) trackways, most of the variation is explained by the first principal component. For the ornithopod (I1) trackway, on the other hand, variability is more equally dispersed amongst the principal components as no clear variation patterns are discernible.

Figure 12 Warped outline drawings illustrating shape changes described by the principal components (red outlines, solid dots) relative to the mean shapes (blue outlines, hollow dots) in positive direction (scale factor: 0.1).

Note that the principal component analysis was based only on the six landmarks (red and blue dots), and that the warped outline drawing connecting the dots is shown only for visualization purposes. For T2, only the first two principal components are taken into account as the third principal component only accounts for 7% of the total variation and forms a plateau with the remaining PCs, thus being uninformative.

Further statistical analyses are based only on the six landmark points, while semi-landmarks are not used. Direct comparison of landmark variances confirms observations made based on the vector plots (Fig. 13). In the T3 outlines, variance is most unevenly distributed among the landmarks, with major variation occurring in the lateral hypex, and, to a lesser degree, in the heel and in the medial hypex. In the T2 landmarks, variance is highest in the lateral hypex and the heel, while the variance in the medial hypex is on par with the variance of landmarks 3 and 5. Amongst the I1 landmarks, variance is more evenly distributed, with the greatest variance seen in the lateral hypex and the lowest variance seen in the middle digit. The variance in the heel is a bit below the average variance of all six landmarks.

Figure 13 Bar chart showing the variance of the individual landmarks for the I1, T2, and T3 trackways.

In contrast to the I1 trackway, the T2 and T3 trackways show a strikingly high variability in landmark 2 (lateral hypex) and 6 (heel).

The principal component analysis of the T3 outlines reveals a high loading on the first principal component, which describes 46% of the total variance (Fig. 11). The main shape change in PC1 occurs in the lateral hypex (landmark 2), which is shifted posteriorly relative to the mean shape, and in the heel point (landmark 6), which is shifted anteriorly (Fig. 12). PC2, accounting for 23% of the total variance, mainly describes the variation in the medial hypex, which is shifted anteriorly compared to the mean shape. PC3, describing 17% of the variance, describes a somewhat reduced digit divarication. For the T2 outlines, PC1 accounts for 55% and PC2 for 32% of the total variance. Thus, 87% of the total variance can be described by the first two PCs (Fig. 11). In PC1, the lateral hypex is shifted posteriorly while the heel point is shifted anteriorly, closely resembling PC1 of the T3 outlines (Fig. 12). PC2 shows shape changes in the tips of digits II and III. The remaining PCs were disregarded as they slowly level out, forming a plateau. The I1 outlines show low loadings on the first PCs, with 28% of the total variation described by PC1, 26% by PC2 and 19% by PC3 (Fig. 11). Variation patterns are less clear, with the main variation of the heel point being described by PC1, the main variation in the lateral hypex described by PC2 and the main variation of the medial hypex described by PC3 (Fig. 12).

Variability of trackway parameters

The analyzed section of the T3 trackway (T3/23–T3/48, Fig. 2) starts with a broad right turn, where it crosses the I1 trackway twice. After the second crossing, the direction of the trackway remains constant, although its course is sigmoidal. At the second crossing, footprint T3/33 appears to be stepped over I1/34, suggesting that the T3 trackway was made after the I1 trackway. The best preserved footprint (the cast of T3/18) measures 40.4 cm in length, including the metatarsophalangeal pad, translating into a hip height of 162 cm and a body length of 426 cm. The average pace length is 115 cm, and the average stride length 228 cm. Pace angulation amounts to 163° and footprint rotation to 1.7° on average. The average speed equals 1.77 m/s (6.36 km/h), with a maximum value of 1.93 m/s (6.96 km/h).

The T2 trackway (T2/1–T2/24, Fig. 2) is more straight than the T3 trackway, only showing a very slight tendency towards the left during its course. It runs sub-parallel to the distal two thirds of the T3 trackway. T2/18 is missing, being overstepped by the crossing I1 trackway, indicating that the T2 trackway already existed when the I1 trackmaker left its track. Thus, the T2 trackmaker probably was the first of the three animals to cross the surface. Pace lengths average at 104 cm and stride lengths at 208 cm. Footprint rotation is very weak and close to 0° on average. Foot placement in the T2 trackway is remarkable, as 12 of 19 footprints are located medial to the trackway midline, a condition henceforth called cross-over gait following McClay & Cavanagh (1994). The pace angulation averages at 183°, with a maximum value as high as 193°. Most footprints lack a fully impressed heel. The completely preserved mold of T2/22 measures 29.5 cm in length, indicating a hip height of 118 cm and a body length of 311 cm. The calculated speed averages at 2.19 m/s (7.91 km/h), with a maximum value of 2.27 m/s (8.18 km/h).

The analyzed part of the I1 trackway (I1/17–I1/53) crosses the T3 trackway two times (at I1/19 and I1/34). Between the two crossings, it describes a turn to the left, after which it is continuing rather straight, crossing the T2 trackway at I1/45. As indicated by the oversteppings at the crossings, the I1 trackmaker probably crossed the surface after the T2 trackmaker, but before the T3 trackmaker. The mean footprint length of the selected footprints is 29.5 cm, suggesting a hip height of 118 cm. Pace length accounts for 66 cm and stride length for 131 cm on average. Pace angulation equals 167°, and footprint rotation −11.2°, indicating pronounced inward rotation. The speed averages at 1.02 m/s (3.66 km/h); thus, the ornithopod was progressing significantly slower than both theropods.

All three Münchehagen trackways show relatively constant locomotion speeds; the difference between maximum and minimum values accounts for 0.18 m/s in the T2, 0.31 m/s in the I1, and 0.44 m/s in the T3 trackway. Pace lengths are most variable in the I1 trackway, with a coefficient of variation (CV) of 5.70, followed by the T3 (4.13) and the T2 trackway (2.88). Likewise, stride lengths vary with a CV of 4.39 in the I1, 3.86 in the T3 and 1.76 in the T2 trackway. Variation of pace angulation is greatest in the T3 trackway, with a standard deviation (SD) of 10.78, followed by the I1 (8.00) and T2 (5.94) trackways. Footprint rotation is slightly more variable in the I1 (SD =8.83) than in the T3 (SD =8.70) trackways; in the T2 trackway, SD is only 5.95.

Discussion

Objective definition of footprint margins: comparison of methods

The definition of the margin of a footprint constitutes a complex three-dimensional problem. Three-dimensional surfaces can be reduced to contour maps, transforming the three-dimensional problem into an easier-to-handle two-dimensional one, greatly facilitating the definition and application of specific criteria. Although the interpretational approach employed here captured the foot anatomy more faithfully than the tested objective approaches, it may involve a problematical degree of subjectivity. Thus, results of any subsequent quantitative analysis can not be fully objective (Falkingham, in press). The development of objective approaches for the definition of footprint extents is therefore of urgent importance for the quantitative study of fossil footprints.

Working with contour maps, the most straightforward approach is the selection of a single contour line, in order to define the footprint boundary on a constant height level (e.g., Romilio & Salisbury, 2014). Testing this approach on the Münchehagen footprints revealed important shortcomings. Since contour lines at different depths may differ considerably in shape within a single footprint, the necessary selection of a single contour line still introduces a significant amount of subjectivity (Falkingham, in press). Furthermore, a single contour line is strongly affected by noise and extramorphological influences, and often cannot depict a footprint in its full extents. While an interpretative outline aims to capture the important features of the footprint wall, a single contour line can only represent a much less informative, arbitrary representation of the footprint, as features outside of the height level of the contour line are ignored. These problems might be partly solved by calculating the mean shape of all contour lines describing the footprint wall using GPA. In practice, however, the height of the footprint wall varies within the footprint. The stack of contour lines that can be analyzed will therefore be restricted by the shallowest part of the footprint wall, so that deeper parts are only partly covered by the analyzed stack of contour lines.

A different approach, the consistent selection of the steepest slope, is associated with different practical problems. First, the steepest slope does not necessarily represent the margin of the actual footprint stamp. The steepest slope can occur both on the outer area of the footprint close to the border to the undeformed sediment, and inside the footprint, e.g., when the distal part of a digit impression is partly filled with sediment, forming a steep slope at the base of the infill. In both cases, the steepest slope will convey only little information on the actual foot anatomy. This problem is most evident in several of the I1 footprints. Second, the steepest slope can rarely be followed along the whole outline; rather, it fades out frequently, causing the outline tracing to abruptly jump to a different height level. The latter problem might be solved by detecting the steepest slope on multiple points along the footprint margin, and approximating a single outline using splines or elliptic Fourier transforms.

Objective landmark positions resulting from the contour line approach were subjected to GPA and principal component analysis for all three trackways, reproducing the results given by the same analysis based on interpretative landmarks. Interpretative bias therefore cannot explain the observed variation patterns.

Causes of variability

In theory, footprint shape is determined by three factors, namely the anatomy of the trackmaker’s foot, substrate properties, and behavior (Falkingham, 2014). Additional factors affecting footprint shape include pre-burial and recent alteration (e.g., Henderson, 2006; Scott, Renaut & Owen, 2010) as well as diagenesis (e.g., Lockley, 1999; Schulp, 2002; Lockley & Xing, 2015). The tracks were documented shortly after excavation, limiting exposure to the elements. The removal of the overburden using excavators frequently damaged the brittle tracking layer, contributing to the observed variability. However, digital comparisons of three footprint negatives with their casts (T3/44, T3/45 and T3/46) showed that shape differences due to material loss during excavation were minimal at least in these examples (Wings, Lallensack & Mallison, in press). Last but not least, subjectivity and noise introduced by interpreting the footprint outlines and determining the landmark points will inevitably contribute to the observed variability. Although subjectivity was reduced by applying specific criteria for the tracing of outlines, these criteria are not always applicable unambiguously. As discussed in ‘Objective methods for the definition of footprint margins’, a second geometric morphometric analysis using landmark positions derived from an objective approach was able to reproduce the observed patterns at least for the T3 and T2 trackways (Figs. S1 and S2), suggesting that interpretational bias, although considerably contributing to the observed variability, cannot explain the observed variation patterns.

A significant difference between left and right footprints was not observed in any of the three trackways. As foot anatomy does not change within a trackway, the substrate properties and the behavior are presumably the major causes of variability within the present trackways. Whether substrate or behavior is the major contributor to variability is generally difficult to assess, and depends on the footprint feature or measurement in question. Both theropod trackways differ from the ornithopod trackway in the pronounced variation patterns in the heel and lateral hypex areas. It is unclear whether these differences might be the result of locomotory differences in separate individuals, or even separate trackmaker groups such as ornithopods and theropods (Lallensack, 2015). Alternatively, the differences might be explained by the presumably higher body weight of the ornithopod, which might have resulted in a more regular impression of the heel and hypex areas. However, as the absence of the distinctive variation pattern in the I1 trackway could also be a random effect given the small sample size, further research is needed to investigate these possibilities.

Variability of quantitative and qualitative track features

A wide array of quantitative and qualitative track features have been employed for the characterization of tracks (Lockley, 1998). The trackway pattern can be completely characterized by quantities, i.e., linear measurements such as pace- and stride lengths, and angular measurements such as pace angulation and footprint rotation. A comprehensive characterization of individual footprints encompasses both quantities and qualities (Lockley, 1998). The former include the number of digits and digital pads, linear measurements, such as the dimensions of the overall footprint and those of the individual digits and pads, and angular measurements, most importantly the interdigital angles. Qualities can include the shape, relative position, and orientation of parts of the footprint such as ungual impressions, the heel region, the hypices, or the pad impressions.

Linear measurements include both information on shape and size. Ichnotaxonomically meaningful comparisons are only possible when the influence of size is minimized, e.g., by using ratios. Nevertheless, mere size, usually approximated by footprint length, is commonly employed to distinguish ichnotaxa. For example, the ubiquitous ichnogenus Grallator is restricted to footprints less then 15 cm in length (Olsen, Smith & McDonald, 1998), while the newly described ichnogenus Bellatoripes was diagnosed to encompass footprints over 50 cm in length (McCrea et al., 2014). Other diagnoses make use of more general categories, such as “small size,” “medium size” and “large size” (e.g., Xing et al., 2013; Xing et al., 2014a; Lockley, Meyer & Dos Santos, 1998). Employing size for the diagnosis of ichnotaxa appears questionable, as such categories are necessarily arbitrary. Such an approach can lead to an overestimate of the diversity present in a sample, as different ontogenetic stages of the same species would fall into separate ichnotaxa (Bertling et al., 2006). This is especially problematic since size difference between hatchlings and adult individuals is large, especially in larger dinosaurs.

Below, we review a selection of commonly employed qualitative and quantitative track features, and discuss their intratrackway variability based on the findings derived from the Münchehagen trackways.

Hypex positions and associated measurements. For excellently preserved footprints, hypices are rarely used as descriptors, since reference points based on well defined pad impressions are much more informative. Hypices become increasingly important for less well preserved material lacking discrete pad impressions, as useful reference points in such footprints are scarce. In the measurement scheme proposed by Moratalla, Sanz & Jimenez (1988), which does not require the presence of pad impressions, 11 of the 18 measurements directly depend on the medial or lateral hypex (Fig. 8B). This includes commonly employed measurements such as the free lengths of digit impressions II, III, and IV, the heel-interdigital distances, and the digital widths, which also are important parameters in the discrimination of theropod and ornithopod footprints (e.g., Moratalla, Sanz & Jimenez, 1988). A qualitative feature, the relative position of the two hypices, is occasionally used to define new ichnotaxa (Lockley et al., 2006; Lockley et al., 2007; Xing et al., 2014c).

Belvedere (2008) observed that the hypices are the most variable of the six analyzed landmark positions in a theropod trackway from the Late Jurassic of Morocco, concluding that the hypices in general should not be used as features in ichnotaxonomy. In the Münchehagen footprints, the lateral hypex was determined the most variable of the six defined landmark positions in all three trackways (Fig. 13). Variability of the medial hypex is significantly lower, although still representing the second most variable landmark in the I1 and the third most variable landmark in the T3 trackway. The overall increased variability of the hypex landmarks is in accordance with the findings of Belvedere (2008). In the Moroccan trackway, however, the medial hypex was found to be more variable than the lateral hypex (Belvedere, 2008), contrary to the condition in the Münchehagen tracks.

Intriguingly, the main variation in the medial and lateral hypex points appears in separate principal components in all three trackways, indicating that the lateral and medial hypex positions vary independently from each other. This suggests that the relative position of the hypices is a potentially very variable feature and is only informative if large sample sizes are available. Likewise, any measurements depending on the hypex positions should be used with caution. The free length of digit III can be determined by taking into account both the medial and lateral hypex, reducing extramorphological influences. On the other hand, measurements of the free lengths of digit impressions II and IV are necessarily based on only one of the two hypex positions, diminishing their informative value.

High variability in hypex positions in theropod footprints might result from different factors. First, hypices are non-compressed areas and as such are likely to be more influenced by variations caused by foot-sediment interactions than the highly loaded digit impressions (Belvedere, 2008). Furthermore, hypices can be expected to be influenced by trackmaker behavior, e.g., through changes in the interdigital angle and the degree to which the posterior part of the digits are impressed. Second, hypices are strongly influenced by preburial and recent erosion, especially when the interdigital angle is low, as the narrow sediment rims between the digit impressions are the first features to be eroded (Henderson, 2006). On the other hand, a preservation as undertracks less likely affects hypex positions according to Henderson (2006). Last but not least, inferred hypex positions can very much vary when interpreted by separate researchers. In many of the Münchehagen footprints, the posterior end of the sediment bar separating the digit impressions fades out indistinctly into the base of the footprint, without showing a single distinct slope, making their identification highly subjective. As the result of an experiment, Thulborn (1990) illustrated eight different outline drawings drawn by separate persons based on the same photograph of a theropod footprint, in order to illustrate the influence of personal interpretation. Our examination of the outline drawings shows that in four interpretations the hypex of the right side is located posterior to the hypex on the left side, while the relative hypex positions are vice versa in two and equal in yet another two interpretations. As a conclusion, hypex positions in published outline drawings are probably not informative in most cases.

Heel region and associated measurements. The extension and morphology of the heel region is frequently employed in ichnotaxonomy (e.g., Langston Jr, 1974; Gangloff, May & Storer, 2004; Lockley et al., 2014; Xing et al., 2014a; Díaz-Martínez et al., 2015), as well as for discriminating between theropod and ornithopod (e.g., Moratalla, Sanz & Jimenez, 1988; Pittman, 1989; Thulborn, 1990; Mateus & Milán, 2008; Xing et al., 2014b) and between left and right footprints (e.g., Pittman, 1989; Thulborn, 1998; Marzola & Dalla Vecchia, 2014). Several common measurements depend on this feature (Fig. 8B), most importantly the footprint and digit impression lengths. Footprint length is of crucial importance not only for describing overall footprint dimensions, but also for estimating hip height and locomotion speed of the trackmaker and associated paleobiological inferences (Falkingham, in press).

In both theropod trackways, the antero-posterior variation of the landmark on the heel constitutes the second largest “hotspot” of variability, only excelled by the landmark on the lateral hypex, which according to PC1 covaries with the heel landmark (Fig. 12). This covariation might be explained by variations in substrate properties or erosion. Demathieu (1990) suggested that the shape of the heel depends on the sinking depth of the foot, and thus on the sediment properties. This is not evident in the Münchehagen trackways, as PC1 does not significantly correlate with the maximum depth of the footprints. The hypex positions might be highly susceptible to changing substrate properties (Belvedere, 2008) or erosion (Henderson, 2006). However, in the T3 mean shape, the interdigital angle between digit impressions III and IV is larger than that between digit impressions II and III, resulting in a larger interdigital sediment bar that is less likely to be partially erased. The consistently higher variability in the lateral hypex thus appears counter-intuitive.

As an alternative explanation, PC1 might reflect behavioral differences of the animal, caused by variations in the degree to which the proximal part of digit IV was impressed. In recent ostrich (Struthio camelus) footprints, the presence of metatarsal impressions was suggested to be at least partly determined by behavior (Belvedere & Mallison, 2014), opening the possibility that the same holds true for tridactyl dinosaur footprints. Variations in the impression of the heel in a large tridactyl trackway from the Australian Lark Quarry have been suggested to result from different pedal postures, and thus, behavior (Thulborn & Wade, 1984; Thulborn, 2013; Romilio & Salisbury, 2014). Furthermore, it has been suggested that the degree to which the heel is impressed can vary with locomotion speed (and thus behavior), with running animals impressing only the distal parts of their digits (Sarjeant, 1975; Thulborn & Wade, 1984; Lockley & Conrad, 1989). However, foreshortened digit impressions and long stride lengths can in some cases be interpreted as swimming tracks (Romilio, Tucker & Salisbury, 2013). Contrary to the theropod outlines, the heel extent in the I1 outlines does not show an strongly increased variability, indicating a more constant impression of the heel pad.

The extent of the heel impression can vary not only due to an incomplete impression of the foot, but also due to the additional impression of the metatarsus. Although not preserved in the Münchehagen tracks, metatarsal traces are frequently found in footprints attributed to both theropods and ornithopods, and can be caused either by behavior or sinking depth of the foot into the sediment (e.g., Kuban, 1989; Citton et al., 2015; Lallensack et al., 2015; Pérez-Lorente, 2015). Several ichnotaxa are based on material including metatarsal impressions, causing ichnotaxonomical problems (e.g., Díaz-Martínez et al., 2015). In the light of the potential high variability in the extension and morphology, features and measurements related to the heel region should only be used when a full impression of the foot can be ascertained and a contribution of the metatarsus can be excluded.

The analyzed theropod footprints do not only show anteroposterior, but also mediolateral variation in the degree to which the heel is impressed. Thus, digit IV can be fully impressed while large parts of the proximal portion of digit II are not impressed, and vice versa. This results in a spectrum of different morphologies, including V-shaped, U-shaped, symmetrically bilobed and asymmetrically bilobed shapes. The I1 trackway, although showing a more constant heel pad impression, shows various different heel morphologies ranging from V-shaped or broadly arched to sub-rectangular; both width and position of the heel pad relative to digit impression III varies greatly. The consequences of the high variability in the heel on the differentiation between left and right footprints and on trackmaker identification are discussed below.

Morphology and dimensions of digit impressions. Morphology and dimensions of digit impressions are important characters in ichnotaxonomy (e.g., Lockley, 1998; Lockley, 2009; Lockley et al., 2014) as well as in the distinction between left and right footprints (e.g., Pittman, 1989; Thulborn, 1990) and between theropod and ornithopod footprints (e.g., Farlow, 1987; Moratalla, Sanz & Jimenez, 1988; Thulborn, 1990; Romilio & Salisbury, 2011; Thulborn, 2013; Romilio & Salisbury, 2014). Dimensions of the digit impressions can be described using the length-to-width ratio (e.g., Moratalla, Sanz & Jimenez, 1988), or by assessing the relative digit lengths, such as the projection of digit III beyond digits II and IV (e.g., Lockley, 2009), or the projection of digit IV relative to digit II (e.g., Xing et al., 2014d). Informative qualitative features include the tips of the digits, which may record the presence of claws or hooves.

In the T2 trackway, the tip of digit impression II generally extends beyond that of digit impression IV, as indicated by both the mean shape and well preserved footprints (Figs. 3 and 6). Landmark analysis indicates a slightly increased variability on digit impression II (Fig. 13). Foster (2015) discarded the relative extension of digits II and IV as an informative feature for Hispanosauropus tracks from the USA and Spain, as this feature was found to be very variable in these tracks.

Measurements of the length-to-width ratio can be problematic when digital pads are not well defined, as the variability in the heel (for measurements of overall digit lengths) and the hypices (for measurements of the digit free lengths) is difficult to assess. Furthermore, digit impression width varies greatly in the described footprints, especially in digit impression III. In the T2 trackway, the marked narrowness seen in the digit impressions of some footprints is possibly due to sediment being drawn inside the toe impression when the toe was withdrawn (Thulborn, 1990; Gatesy et al., 1999). A high degree of variability in the width of the digit impressions has been noted for other tracksites as well (e.g., Farlow et al., 2007).

Digit impression morphology remains relatively constant within both theropod trackways. The morphology of the distal tip is determined by the position of the claw. The mean shapes and best preserved footprints of both trackways indicate a central position of the claw of digit impression II, a medial position in digit impression III, and a lateral position in digit impression IV. Claw locations frequently deviate from this general position even in excellently preserved footprints, indicating behavioral variability. High variability in claw positions is a general feature in many theropod dinosaur footprints (Thulborn, 1990).

In the I1 trackway, digit impression morphology is more variable than in the theropod trackways; while appearing long and narrow in some examples, they approach a cloverleaf-shape in others. Digit impressions II and IV appear straight in some footprints, but show a bend at midlength in others, producing a pronounced inwards curvature. The foot anatomy in ornithopods generally allows some degree of mediolateral bending in digits II and IV—the combination of deformations in the digital joints and the soft parts during footprint formation might result in the observed morphology (T Hübner, pers. comm., 2016). Curvature of digit impressions can occur as a result of foot-substrate interactions even when the digits themselves are straight, as was recently demonstrated with computer simulated footprint formations, although at a much smaller scale (Falkingham & Gatesy, 2014). In conclusion, it is not clear to which degree the observed curvature reflects anatomical features. In both theropod trackways, digit impressions are generally straight, with the exception of digit impression IV in T3/47 (Fig. 5M), which shows a lateral bend. This bend is most pronounced in the distal third of the digit impression, where it probably results from a lateral orientation of the claw (cf. Thulborn, 1990).

Trackway parameters. Trackway parameters are commonly employed in ichnotaxonomy (e.g., Lockley et al., 2014; Lockley, Meyer & Dos Santos, 1998), despite the potential high influence of behavioral variability (Díaz-Martínez et al., 2015). They are also used to distinguish theropod and ornithopod trackmakers, with ornithopods tending to show shorter pace- and stride lengths, a lower pace angulation, and a stronger inward rotation (Lockley, 1987; Farlow et al., 2007; Castanera et al., 2013). Furthermore, the trackway pattern represents the most obvious criterion for the distinction between left and right footprints. All variables of the trackway configuration, regardless if describing the distance between footprints (pace and stride lengths), the width of trackway (e.g., pace angulation), or the pes rotation, are strongly influenced by the locomotion speed of the trackmaker (cf. Alexander, 1976; Day et al., 2004; Kim & Huh, 2010). As can be expected for an ornithopod, the I1 trackway shows shorter pace- and stride lengths and a higher pes rotation than both theropod trackways. However, pace angulation is slightly higher in the ornithopod trackway (167° on average) than in the T3 theropod trackway (165° on average); this parameter therefore cannot unambiguously differentiate between theropod and ornithopod footprints in the present tracks.

Stride lengths (and, consequently locomotion speeds) are relatively constant within all three trackways. The maximum locomotion speed of the T3 trackmaker of 1.93 m/s is in accordance with the independent estimate of 6.5 km/h (1.81 m/s) proposed by Troelsen (2015). Our estimate for the T2 trackmaker (2.27 m/s or 8.18 km/h) however is lower than that of the latter study (12 km/h). As a whole, the T2 trackway appears very straight and regular, with only a slight bend to the left. The I1 trackway, apart from the abrupt turn to the left at footprint I1/29, is also reasonably straight and slightly sigmoidal. The T3 trackway, on the other hand, is more strongly sigmoidal, although its general course remains constant after the turn to the right between footprints T3/26 and T3/30. Similar sigmoidal trackways have been described by several authors for both theropod and ornithopod trackmakers (see Pérez-Lorente, 2015 and references therein). The T2 trackway shows the smallest variability in all measured trackway parameters, possibly due to its straight course and higher locomotion speed. Variability of the pace lengths is greater than that of the stride lengths in all three trackways.

For measuring trackway parameters, most studies utilize the tip of digit impression III as a corresponding reference point (Thulborn, 1990). Alternatively, the base of digit impression III might be used for this purpose (Thulborn, 1990). The divergence of results by the two approaches usually is negligible for trackways with long strides, as has been suggested by Farlow (1989) based on extant ostrich (Struthio camelus) trackways. The Münchehagen trackways, however, show a strong variability in footprint rotation, and, in case of the I1 trackway, a strong inward rotation, possibly significantly influencing results. Reference points on the tip of digit impression III give generally higher standard deviations than those on the base of that digit impression for all trackway parameters (Table S1). This suggests that measurements based on the digital bases are somewhat more informative since the influence of footprint rotation variability is reduced. Despite marked differences in standard deviation especially in the T3 trackway, average values derived from both approaches are very similar for most trackway parameters (Table S1). Average pace angulation in the I1 trackway is an exception, being increased by 10% when the tip rather than the base of digit impression III is used, a result of the strong inward rotation of the footprints in this trackway.

To our knowledge, systematic cross-over gait along most of the trackway course has not been reported in any other dinosaur trackway. When present, it is usually restricted to one single step, most frequently during turns (Xing et al., 2014b). A large tridactyl trackway from the Lark Quarry of the Upper Cretaceous of Queensland, Australia, shows a crossing of the trackway midline in probably four out of nine footprints, including two successive footprints with cross-over gait (Thulborn & Wade, 1984; Romilio & Salisbury, 2014). The reason for the observed cross-over gait in the T2 trackway is unclear, as muscular requirements caused by a mediolateral shift in the center of mass can be expected to be higher compared to a foot placement directly on the trackway midline (cf. McClay & Cavanagh, 1994). Given the apparent rarity of such a feature in dinosaur trackways, a biomechanical reason appears unlikely; rather, the cross-over gait might reflect peculiar behavior of the individual. A pathological explanation also cannot be ruled out. Razzolini et al. (2016) recently described an abnormal gait in an ornithopod trackway from the Lower Cretaceous of Spain, which can be attributed to a pathology in the pes of the animal recorded in the footprint morphology. In the T2 trackway, a statistically significant left–right asymmetry could not be detected neither in the footprint morphology nor in the trackway parameters.

Distinction between left and right footprints

The correct identification of a footprint as pertaining to the left or the right foot is crucial when descriptors related to footprint asymmetry are employed, such as any differences in divarication, morphology, and size between digit impression II and IV, relative orientation of digit- or heel-impressions, curvature of digit impressions, and orientation of the ungual impression, amongst others. The most straightforward criterion is the position of the footprint relative to the trackway midline – this criterion, however, is not infallible even when long trackways are available, as shown below. Theropod tracks often show a pronounced asymmetry, allowing the assignment to the left or right foot even based on isolated footprints, while larger ornithopod tracks usually are subsymmetrical (Díaz-Martínez et al., 2015, but see below). For theropod footprints, left–right criteria include the configuration of the heel region, the curvature of digits, and the orientation of claw marks. Additional criteria might be employed occasionally, such as smear marks originating from the tips of the digit impressions (Thulborn, 1998), pressure release structures between digit impressions (Martin et al., 2012), or, if metatarsal impressions are present, the location of the hallux impression as well as the angling of the acropodial against the metapodial impression (Pérez-Lorente, 1993). In bird footprints, interdigital angles can be used as yet another criterion, as the angle between digits III and IV is often wider than between digits II and III (Padian & Olsen, 1989). This configuration appears to apply to many theropod footprints as well (Farlow 1987; Pérez-Lorente 2015, but see Thulborn 1990 for a contrary statement).

Few attempts have been made to differentiate left and right footprints of larger ornithopods based on pes morphology (for exceptions, see e.g., Currie, Nadon & Lockley, 1991; Hornung et al., 2012), as such tracks generally display a high degree of symmetry (Díaz-Martínez et al., 2015). Contrary to this assumption, the mean shape of the I1 ornithopod footprint shows the subcircular heel pad being clearly displaced laterally with respect to digit impression III. Footprint asymmetry is highlighted by GPA-based comparisons of the T3, T2 and I1 mean shapes with their respective mirror images (Fig. 9D). Asymmetry is most pronounced in the T3 mean shape, with a Procrustes distance of 0.0089 between the mean shape and its mirror image, followed by the T2 mean shape (0.0045). Procrustes distance in the I1 mean shape is only slightly smaller, accounting for 0.0034. Below we test the three most important traditional criteria for the distinction between left and right footprints—the location of the footprint relative to the trackway midline, the heel configuration, and the orientation of digit III.

An asymmetric distal end of digit III can be seen in both theropod trackways, while in the ornithopod trackway digit III is symmetrical and thus not informative. Both the curvature and claw mark of digit III usually point to the inside of the trackway (Pittman, 1989; Thulborn, 1990). In the theropod footprints, asymmetry can result from both the position and orientation of the claw mark, usually towards the medial side. When the claw impression is not distinct, the termination of digit impression III is usually arched on the lateral and straight on the medial side. Of the 23 preserved footprints of the T2 trackway, asymmetry of digit impression III could be observed in 13 footprints. 11 footprints could be correctly classified as pertaining to either the left or right foot, while the classification was in error for 2 footprints (T2/1 and T2/3) in which asymmetry is only weakly expressed. The interpretation of footprint T2/17 is ambiguous, as the well preserved claw impression is located on the lateral side but tilted towards the medial side. In the T3 trackway, asymmetry could be observed in 15 footprints, all of them being classified correctly. In conclusion, the morphology of the distal end of digit III appears to be a reliable left–right criterion for footprints with well-developed asymmetry.

In theropod and smaller ornithopod footprints, digit IV is often impressed along its whole length including the metatarsophalangeal pad, while the proximal parts of digits II and III are held off the ground (Baird, 1957; Farlow et al., 2000). In the footprints, this condition often results in an indentation along the medial side of the footprint separating digit II from the metatarsophalangeal pad of digit IV (Farlow et al., 2000). Both the indentation of the medial side and the proximal extension of digit IV past digit impression II are commonly used to distinguish left and right footprints, even when the phalangeal pads are not visible (Marzola & Dalla Vecchia, 2014; Pittman, 1989; Thulborn, 1998). In the T2 trackway, 8 of the 23 footprints show a clearly asymmetric heel impression (most pronounced in T2/22), and all 8 footprints could be correctly classified as either left or right based on this feature. The T3 footprints, on the other hand, proved to be more ambiguous. In footprints preserving an impression of the metatarsophalangeal pad (best seen in T3/18), the heel varies from being strongly asymmetrical (e.g., T3/47) to being V-shaped, with only a slight tendency towards the lateral side and a quite weakly developed medial notch. The heel is clearly asymmetrical in 12 of the T3 footprints—while our classification was correct for nine of the footprints, it was incorrect for tree examples. All three incorrectly classified examples show a foreshortened digit impression IV being less extensive than digit impression II, resulting from an incomplete impression of the heel. These footprints, however, can be distinguished from examples showing a fully impressed heel in that the proximal ends of digits II and IV are more widely separated from each other, resulting in a much broader heel.

In the I1 trackway, distinction between left and right footprints is possible due to the asymmetric heel region, which proved to be a surprisingly reliable criterion. 18 footprints show a marked asymmetry in the heel, and all but one (I1/42) could be correctly classified as either left or right footprints based on this feature. A similar asymmetry appears to be present in several other ornithopod tracks from the Obernkirchen Sandstone (cf. Lockley & Wright 2001, Fig. 29.1B; cf. Hornung et al. 2012). The observed asymmetry probably results from an asymmetry in the foot anatomy of the trackmaker, and is possibly homologous to the asymmetric condition in theropods, which seems to represent the basal condition in dinosaurs (Farlow et al., 2000). Although metatarsal traces are frequently reported for ornithopod tracks (Lockley, Young & Carpenter, 1983; Pérez-Lorente, 1993; Loza, Medrano & Lorente, 2006; Vela & Lorente, 2006; Lucas et al., 2011; Pérez-Lorente, 2015), such traces tend to be angled medially with respect to the foot’s long axis, contributing to the marked inward rotation of the foot (Pérez-Lorente, 1993; Pérez-Lorente, 2015). As the asymmetric heel pad is located laterally with respect to the foot’s long axis, a significant contribution of the metatarsal shaft to the observed asymmetry appears unlikely.

Identification of left and right footprints based on their position relative to the trackway midline was expectedly unambiguous for the T3 and I1 footprints, but misleading for the T2 footprints. In the T3 trackway, all 20 measurable footprints are located on the expected site of the trackway, with only T3/39 being on-line with the preceding and subsequent footprint. The same holds true when the tip of digit impression III is chosen as the reference point. Likewise, in the I1 trackway, all 26 measured footprints fall on the expected side of the trackway midline, when the base of digit impression III is taken as the reference point. However, when the tip of digit impression III is chosen as the reference point, 7 of the 26 measured footprints are located medial to the trackway midline, with an additional 5 being located on the trackway midline. This apparent overstepping results from the inward rotation of the footprints. In the T2 trackway, 12 of the 19 measured footprints show pronounced overstepping over the trackway midline, with only 6 footprints being located on the expected side and one directly on the midline. With the tip of digit impression III as the reference point, only five footprints fall on the expected side of the trackway, with two located directly on the trackway midline. This observation contradicts all other left–right criteria, including the heel configuration and the orientation and position of the claw of digit III. Examples with preserved phalangeal pads, such as T2/22 (Fig. 6G), confirm the presence of cross-over gait in most of the trackway.

Implications for the discrimination between ornithopod and theropod footprints

The distinction between ornithopod and theropod footprints can be difficult even when based on complete trackways (Lockley, Foster & Hunt, 1998; Farlow et al., 2007; Romilio & Salisbury, 2011; Thulborn, 2013; Romilio & Salisbury, 2014; Hübner, in press). The general appearance of the I1 differs considerably from the T2 and T3 mean shapes, most obviously in the heel region. However, direct comparison of the mean shapes reveals striking similarities in proportions (Figs. 9B and 9C), indicating that most traditional measurements might not be able to discriminate between footprints of both trackways. Important commonly employed criteria include the length-width ratio, assuming that theropod footprints in general are longer than wide, while ornithopod footprints are as wide as long or even wider (Moratalla, Sanz & Jimenez, 1988; Thulborn, 1990; Farlow et al., 2007; Romilio & Salisbury, 2014). Strikingly, in both the T3 and I1 mean shape, the length-width ratio is 0.91; the footprints thus are markedly wider than long, while the T2 mean shape is about as wide as long. Therefore, the length-width ratio is a misleading criterion for both the T2 and T3 mean shapes. A low length-width ratio appears to be not as uncommon in large theropod footprints as previously thought (Lallensack et al., 2015). While a high length-width ratio may still represent a reliable indicator for theropod footprints, the reverse, a low length-width ratio, might not be as reliable an indicator of ornithopod footprints as previously assumed.

Another criterion, the width of the digit impressions, assumes that digit impressions of theropod footprints tend to be narrower than those of ornithopods (Moratalla, Sanz & Jimenez, 1988; Thulborn, 1990; Farlow et al., 2007). In the present mean shapes, the relative widths of digit impressions II and IV are about equal in the T3 and I1 mean shape, with the impression of digit III even appearing wider in the T3 mean shape (Fig. 14B). Only digit impressions III and IV of the T2 mean shape show a reduced width when compared with the T3 and I1 mean shapes. Furthermore, as already discussed, digit proportions are amongst the most variable footprint features, especially in the I1 trackway, ranging from short and wide to long and narrow (Figs. 5–7), indicating that they do not fully correlate with the trackmaker’s anatomy. A striking example of intratrackway variability of digit impression shape can be found in the Upper Jurassic Barkhausen tracksite, where a tridactyl trackway shifts from a theropod-like to an ornithopod-like morphology along its course (Lallensack et al., 2015).

Figure 14 Multivariate analysis of the T3, T2 and I1 mean shapes, adopting the approach of Moratalla, Sanz and Jimenez (1988) for the discrimination between ornithopod and theropod footprints.

Red squares represent the T3 trackway (large theropod), blue circles the T2 trackway (mid-sized theropod) and the green triangles the I1 trackway (ornithopod). A clear separation between the theropod and the ornithopod trackways is not possible in this case. The graphical depiction follows Romilio and Salisbury (2011) and Schulp and Al-Wosabi (2012). Measurements are indicated in Fig. 8.

Other criteria for the distinction of ornithopod and theropod footprints include the shape of the digit terminations (Thulborn, 1990). Theropod footprints often show V-shaped terminations, while the terminations of ornithopod footprints are more U-shaped in outline; these differences are best seen in digit III (Thulborn, 1990). This feature is pronounced in many of the footprints (e.g., the cast of T3/18, and I1/36; Figs. 3B and 7F). In other footprints, ungual marks are absent due to poor preservation. Thulborn (1990) noted that the digit impression III of theropods is sometimes distinctly curved, while that of ornithopods is straight. Digit impression III is straight in all three mean shapes. An informative distinguishing criterion in the present tracks is the asymmetry of the digit impressions. In the theropod footprints, the tips of digit impressions II and IV tend to point towards the outside (away from the footprint midline), due to outwardly directed claw impressions. In the ornithopod footprints, this asymmetry is often reversed, with the tip located more towards the inside of the footprint (Fig. 10). Digit impression III tends to be symmetrical in the ornithopod footprint, but shows a markedly medially displaced tip in the theropod footprint, caused by the medially directed claw. In conclusion, most criteria based on the general shape of the footprints are not able to discriminate the present footprints, and that only anatomical details such as claw impressions and digital pads allow for an unambiguous determination of the trackmaker.

Moratalla, Sanz & Jimenez (1988) presented a multivariate approach to discriminate between theropod and ornithopod footprints. These authors carried out factor and discriminant analyses on 66 footprints previously ascribed to either theropods or ornithopods, in order to estimate the most informative parameters for the discrimination between these groups. From these parameters, nine bivariate ratios were defined. Threshold values were selected for each bivariate ratio, allowing for the classification of unknown material. Although the majority of the analyzed material stems from the Lower Cretaceous of La Rioja, Spain, this approach has recently been applied to classify footprints from different epochs of different parts of the world (Mateus & Milán, 2008; Romilio & Salisbury, 2011; Schulp & Al-Wosabi, 2012; Therrien et al., 2015). Thulborn (2013) recently questioned the use of this approach to classify contentious material, noting that (1) the threshold values are defined subjectively to provide the best separation between point clouds in the bivariate plots, (2) the analysis is essentially a set of bivariate plots and thus not a real multivariate analysis, (3) most of the employed ratios reflect the length-width ratio of the footprint, resulting in a high degree of redundancy, and that (4) all digital axes are required to originate from a single point on the heel outline and thus cannot appropriately describe the footprint shape in many examples.

Before discussing the results of the approach of Moratalla, Sanz & Jimenez (1988) applied to the present mean shapes, we need to point out some practical problems which might affect our results. First, the parameter “basal digit width” of digit impression III (WBIII) is defined as the connection line between the two hypices. Consequently, the value for WBIII will be enlarged when the two hypex positions differ in their anteroposterior position, which is frequently the case especially in theropod footprints (Lockley, 1998), including the T3 footprints analyzed herein. Large WBIII values are considered an ornithopod-like feature by the discriminant analysis. According to both Romilio & Salisbury (2011, Fig. 3B) and Thulborn (2013), the parameter “middle digit width” of digit impression III (WMIII) is measured parallel to WBIII. Again, differences in the relative positions of the hypices will result in a greater WMIII value, causing the bivariate ratio LIII/WBIII to suggest more ornithopod-like proportions. However, Moratalla, Sanz & Jimenez (1988) did not indicate the requirement of WMIII to be measured parallel to WBIII. In the present study, we measured WMIII at the shortest distance across the digit impression, which appears to be the most informative measurement approach. Second, the interpretation of the completed analysis is hampered as discriminant weights are only provided for the individual parameters, not for the ratios. This seems important, as weights for single parameters would be influenced by the size of the individual footprint, while the influence of size is reduced in the ratios. Thus, one cannot assess the relative importance of each of the nine ratios. Furthermore, no discriminating formula was provided, again suggesting that a classification is not possible when the result is not completely unambiguous.

When applied to the T3, T2 and I1 mean shapes, the approach of Moratalla, Sanz & Jimenez (1988) gives inconclusive or even misleading results (Fig. 14). For the T3 mean shape, only two of the nine ratios (L/K and L/M) fall within the theropod field. Of these, L/K plots very close to the threshold value, leaving L/M as the only ratio that unambiguously implies theropod affinities for the T3 trackway. Six ratios fall far inside the ornithopod’s field, while another one (BL3/WMIII) equals the threshold value. In conclusion, the discriminant analysis favors an ornithopod affinity of the T3 trackway. For the T2 trackway, values are generally closer to the theropod’s field. However, five of the nine ratios still indicate ornithopod affinities, two of which are very close to the threshold value. Only the I1 trackway could correctly be classified as ornithopod-like. All nine ratios plot inside the ornithopod’s field, including three that are located very close to the threshold value. The three mean shapes tend to plot together, indicating that the footprint proportions are very similar and that the parameters as defined by Moratalla, Sanz & Jimenez (1988) are, in this case, insufficient to separate the ornithopod from the theropod mean shapes.

The only two ratios indicating theropod affinities of the T3 mean shape (L/K and L/M) include the total footprint length and the heel-interdigital distances. The hypices and the heel are the most variable regions in the T2 and T3 footprints, reducing the information content of the two ratios. This observation was confirmed by a PCA on the Procrustes-fitted landmark coordinates of all three trackways (Fig. 9A). Plotting the first against the second principal component reveals a weak separation of the T2, T3 and I1 shapes. The T3 shapes are best separated from the I1 shapes along the first principal component, which describes a posterior shift of the lateral hypex and an anterior shift in the heel – the very same shape differences are captured by the ratios L/K and L/M. However, both theropod trackways also greatly vary along the first principal component, causing significant overlap with the I1 point cloud. As the I1 shapes are restricted to low scores on the first principal component, footprints with shortened heel regions are unlikely to pertain to the ornithopod. The reverse may well be the case, as the heel regions of several of the T3 shapes are as extensive as in average I1 shapes, causing great overlap. The T2 shapes are best separated, showing high scores on both the first and second principal components. The second principal component shows a more posterior position of the medial hypex and a reduced length-width ratio. Both the T2 and T3 confidence ellipses are elongated, while the I1 ellipse is more circular, indicating that the I1 shapes lack a distinctive variation pattern.

As expected, the separation reached by the CVA is significantly better than that of the PCA, with the T2 trackway being completely separated and a slight overlap between the I1 and T3 trackways (Fig. 9B). The best separation between the ornithopod and the two theropod trackways is reached by CV1. Large values on this axis describe a more posterior position of the hypices, a more anterior position in the heel, and an anteriorly extended digit impression III. CVA, however, is unstable with respect to sampling (Reyment & Savazzi, 1999). As the canonical vectors describe the best possible separation of the three trackways based on our sample, the separation reached by these vectors can be expected to be less clear when additional footprints are incorporated (Reyment & Savazzi, 1999). Consequently, the PCA (Fig. 9A) is the more prudent method to estimate the separation of the trackways.

Potentials of geometric morphometrics for the study of dinosaur footprints

Although the number of studies employing geometric morphometrics on dinosaur footprints increases (Rasskin-Gutman et al., 1997; Rodrigues & Santos, 2002; Belvedere, 2008; Clark & Brett-Surman, 2008; Castanera et al., 2015), the method still is not widely established in this field. In most cases, geometric morphometrics is used to differentiate footprints from different localities, with moderate success. In our study, we propose additional applications both for exploratory and statistical purposes. First, geometric morphometrics proved valuable in comparing different approaches for the definition of footprint margins. Individual tools of this method, such as GPA and elliptic Fourier transforms, may even be used to generate objective outlines. Second, GPA is a valuable tool for the comparison of separate shapes and the quantification of shape differences. In the present study, differences between mean shapes of the analyzed trackways and between left and right footprints are visualized using pair-wise GPA-based comparisons (Fig. 9C and 9D). Procrustes distances allow for the quantification of shape differences between individual shapes. Third, mean shapes are valuable quantifications of the average shape of a sample, as has already been shown for recent human and fossil hominin footprints (e.g., Bennett et al., 2016). Unlike selected single footprints considered as representative for a given sample, a mean shape minimizes the effects of intratrackway variability, thus revealing features that likely reflect the trackmaker’s anatomy. A mean shape will necessarily show less detail than the best-defined examples of the sample. Nevertheless, they may reveal even subtle shape features, such as the slight medial bend in digit impression II or the constrictions that possibly delimit phalangeal pads in the T2 mean shape (Fig. 10B).

Furthermore, geometric morphometrics is able to exactly locate shape differences and variability within the footprint. With traditional linear measurements, it can be difficult to assess the origin of a shape difference, as these measurements depend on at least two reference points, complicating their interpretation (e.g., are proportionally shorter digit impressions caused by a shortening of the digits, an elongation of the heel, a more distal position of the hypices, or a combination of these factors?). Likewise, the coefficient of variation can quantify variability of single measurements (Demathieu, 1987; Demathieu, 1990; Weems, 1992), but the footprint regions responsible for the shape variation are not immediately obvious. With geometric morphometric methods, shapes and mean shapes can be directly compared to visualize even subtle shape differences (Rasskin-Gutman et al. 1997; Fig. 9). This obviates the need for employing ratios of linear measurements, which otherwise would have been necessary to remove the effect of size differences. Variability can be assessed for each landmark position separately (Fig. 10), facilitating the understanding of the mechanisms responsible for the observed variability.

Last but not least, the ability of traditional quantitative measurements to discriminate between ornithopod and theropod footprints is shown to be limited, especially when footprint proportions are similar. Geometric morphometrics is able to capture more information on the footprint shape while limiting redundancy. Semi-landmarks allow to capture qualitative shape features (sensu Lockley, 1998) along the outline, including the heel morphology and the asymmetry of the terminations of the digit impressions, which we have shown to represent the most prominent differences between the analyzed ornithopod and theropod trackways. This reveals room for improvement of existing quantitative methods for the discrimination of trackmaker groups.

Conclusions

The three analyzed tridactyl dinosaur trackways are amongst the longest and best preserved in Germany. The I1 footprints, referable to an ornithopod trackmaker, show narrow digit impressions with symmetrical terminations and a bend at mid-length, and a rounded heel pad that is located lateral with respect to digit impression III. The T3 and T2 trackways can be ascribed to theropod trackmakers based on well preserved claw impressions. They probably represent two separate trackmaker taxa, as indicated by morphological differences evident from the calculated Procrustes mean shapes. The T2 footprints are more gracile than the T3 footprints, showing a narrower digit impression III, a greater projection of digit impression III beyond digit impressions II and IV, no offset in the hypex positions, and a distally elongated digit impression II that is slightly bended medially. In total, at least four trackmaker taxa have been recorded from the Münchehagen tracksite.

Trackway parameters are generally less variable when measured from the base of digit impression III rather than from the tip of that digit, suggesting that the former approach has to be preferred when footprint rotation is strong or variable. Pace lengths are more variable than stride lengths in all three trackways. The T2 trackway shows the least variability in all measured trackway parameters, possibly due to its higher locomotion speed. The T2 trackway is striking in showing cross-over gait along most of the trackway.

All three trackways exhibit a great amount of footprint shape variability. The major causes of variability can be narrowed down to variations in the substrate properties and the behavior of the trackmaker. The two theropod trackways show considerable anteroposterior, but also mediolateral variation in the degree to which the heel is impressed, resulting in a large array of different heel morphologies within the same trackway. Although anteroposterior variability in the extent of the heel is much less pronounced in the ornithopod trackway, heel morphology is likewise variable. Digit impressions tend to retain their general shape but vary in thickness within the two theropod trackways. Digit impression morphology is most variable in the I1 trackway.

Geometric morphometrics proved to be of great value for locating and quantifying shape variability in the footprints. In both theropod trackways, variability of landmarks on the lateral hypex and the heel is markedly increased, while in the I1 trackway variability is more equally distributed amongst the landmarks. Any measurements depending on reference points on the heel or on the hypices should therefore be used with caution. Principal component analysis reveals covariation of separate landmarks. The pattern described by the first principal component is strikingly similar in the two theropod trackways, showing a more posterior position of the lateral hypex co-occurring with a more anterior position of the heel. This pattern might be interpreted to directly result from variations in the degree to which the posterolateral portion of the foot was impressed, although it cannot be excluded that this pattern is mainly the result of varying substrate properties. The anteroposterior positions of the two hypices vary independently from each other within all three trackways, suggesting that the relative hypex positions do not represent an informative feature when the sample size is small.

Given the high degree of interpretative subjectivity introduced during outline tracing, the development of objective means to measure footprint shapes is of crucial importance. Of two a priori defined objective approaches, the steepest slope approach resulted in a lesser variability of landmark positions than the contour line approach, and therefore is less influenced by intratrackway variability. Analyses of landmarks placed using this approach reproduced results derived from landmarks placed on interpretative outlines at least for the T2 and T3 trackways. An interpretational bias, therefore, can be excluded as a probable explanation for the observed variation patterns.

The observed high degree of variability raises the question how strongly criteria commonly employed to differentiate between left and right footprints of theropod trackmakers are affected. Asymmetry in the termination of digit impression III resulting from the position and orientation of the claw is shown to represent a reasonable reliable criterion for both the T3 and T2 trackways. This criterion is found to be misleading for 15% of the T2 footprints, while the T3 footprints could be assigned without any misclassification. Likewise, the heel morphology proved to represent a reliable criterion despite its substantial variability. No misclassifications occurred with the T2 footprints, while 25% of the T3 footprints were incorrectly classified. All incorrectly classified examples possess a much foreshortened and therefore very broad heel. Although ornithopod footprints described in the literature are generally considered to be rather symmetrical, our examination revealed marked asymmetry in the heel region in the I1 trackway. This asymmetry allowed for the correct classification of 17 of the I1 footprints, with only one misclassification, suggesting that assignment of isolated material is possible with a reasonable degree of confidence at least for the present type of ornithopod track. The location of the footprints relative to the trackway midline is an expectedly unambiguous criterion for the T3 and I1 trackways. In the T2 trackway, however, 12 of 19 footprints fall on the other side of the trackway midline due to the pronounced cross-over gait, demonstrating that the relative placement of the footprints is not always an unambiguous left–right criterion.

Footprint shapes of the present theropod and ornithopod trackways, although appearing visually distinct, show strikingly similar proportions. Asymmetry in the terminations of the digit impressions proved to be one of the most informative distinguishing criteria. Applying the multivariate approach of Moratalla, Sanz & Jimenez (1988), the three mean shapes tend to plot together, and only the ratio L/M (footprint length against the lateral heel-interdigital distance) was able to clearly separate the T3 from the I1 shape. The discriminative approach of Moratalla, Sanz & Jimenez (1988) tends to suggest ornithopod affinities for the T3 trackway, and is inconclusive regarding the T2 trackway. According to principal component analysis, the present theropod and ornithopod footprints are indeed best separated by the lateral heel-interdigital distance, although large variability of this parameter in the T3 trackway leads to significant overlap with the I1 footprints. Our results indicate that previous quantitative approaches are not suitable to differentiate the present ornithopod and theropod footprint shapes.

Supplemental Information

Figures S1 and S2 Principal component analysis results of steepest slope approach

Click here for additional data file.

Table S1 Measurements of trackway parameters

Click here for additional data file.

Figure S3 High-resolution version of the sitemap

Sitemap based on photogrammetric data showing the three analyzed trackways (T3, T2, and I1), which represent some of the longest dinosaur trackways from Germany. The proximal sections of the T3 and I1 trackways, excavated before 2011, were not included because photogrammetric documentation is not available. Possible continuations of the T3 and T2 trackways discovered in 2015 are also not included.

Click here for additional data file.

We wish to thank Pernille Troelsen for assistance in separating individual footprints of the photogrammetric models. P. Martin Sander, Sashima Läbe, Michael Buchwitz, and Tom Hübner are thanked for discussions on our technical procedure and interpretations. We furthermore are grateful to Ryan T. Tucker for helpful comments on an early version of the manuscript.

Additional Information and Declarations

Competing Interests

Author Contributions

Data Availability

The authors declare there are no competing interests.

Jens N. Lallensack conceived and designed the experiments, performed the experiments, analyzed the data, contributed reagents/materials/analysis tools, wrote the paper, prepared figures and/or tables, reviewed drafts of the paper.

Anneke H. van Heteren and Oliver Wings conceived and designed the experiments, analyzed the data, contributed reagents/materials/analysis tools, wrote the paper, reviewed drafts of the paper.

The following information was supplied regarding data availability:

Data can be found in Figshare:

10.6084/m9.figshare.3027211

10.6084/m9.figshare.2972329

10.6084/m9.figshare.3025144

10.6084/m9.figshare.3026863

10.6084/m9.figshare.3027067

10.6084/m9.figshare.3027949

10.6084/m9.figshare.3027553

10.6084/m9.figshare.3027385

10.6084/m9.figshare.3029698.

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
