# Peer review of "Geometric morphometric analysis of intratrackway variability: a case study on theropod and ornithopod dinosaur trackways from Münchehagen (Lower Cretaceous, Germany)"

_PeerJ, doi:10.7717/peerj.2059_

## Round 0.1 · original submission · Major Revisions

Both reviewers consider this an important paper using a novel technique, and both agree that it should be published. I agree.

I encourage the authors to take seriously the comments of reviewer 1. I too got lost in the paper, and at times was confused as to what, exactly, you are trying to prove. I am not sure I agree that it needs to be two papers, but I do agree that a large part of the discussion is not exactly pertinent to your introduction and methods. I also agree that your strong initial emphasis on morphometrics gets less importance in the discussion than it should, and is somewhat diminished by the qualitative review.

Both reviewers have some questions as to the analyses, and these should be incorporated into a rework version of the paper.

Reviewer 1 ·

Basic reporting

The quality of written language in the article is very good, however as well as being an extremely long piece, the structure of the work is difficult to follow. This manuscript represents a significant amount of work and really, I think that there are two papers here; by forcing them in to one publication, you lose a good deal of clarity. On the one hand, this paper presents what is (to me) a novel application of geometric morphometrics methods to analyse footprint morphology, and indeed from the abstract through to the end of the methods, that appears to be what we will get. There is brief mention of linear morphometric and trackway measurement parameters here too, but they are not discussed in much detail. There then follows in the results and discussion section, several pages of largely qualitative description of the trackways and prints, interspersed with the odd comment on geometric morphometric results. While there is nothing wrong with recording and publishing these descriptions, it makes it exceptionally difficult to follow the arguments for the application of morphometric methods. For instance, despite there being over a page dedicated to the (lack of) difference between left and right prints in the discussion, at no point in the results is an attempt made to quantify this using morphometrics (although it would be an easy addition to Fig. 9). Furthermore, in the discussion beginning on page 13, there is a four page review of how trackways are traditionally measured sandwiched between discussions of methodological considerations on one side, and applications on the other. The overall impression is that this is a very jumbled piece of writing in sections 4 and 5.

Why not split this in to a review of traditional methods as applied at the Munchehagen site, and a further piece later on geometric morphometrics? I say morphometrics later, as although I can see great potential here, I think there are some methodological shortcomings that need to be addressed before this part of the work is publishable, and that despite the manuscript title, relatively little attention is given to the morphometrics other than its use to identify the mean outline for subsequent traditional description.

Experimental design

The application of geometric morphometrics methods to the anaylsis of trackways is an excellent idea, as it has the potential to remove much subjectivity from ichnological analyses. Both sections of the methods raised some concerns, although I don’t think these will take too long to remedy. In section 3.1 (Data acquisition and geometric morphometrics), the use of photogrammetric methods are well established, and it was nice to have the surfaces of the trackways available to download in the SI. It is unfortunate that the contour colours in Figs. 5-8 represent different depths, as it means that there is no easy way to compare relative depth across the images, even within one figure, especially as the key text is far too small to read. I found myself asking at this point how the outlines were defined and so for clarity, I would swap sections 3.1 and 3.2 around, so you first explain how you defined the footprint outlines before you discuss how you measured those outlines. This is particularly important as a fundamental principle of geometric morphometrics is that the landmarks be homologous between specimens, so reliably defining the outline is absolutely crucial. Assuming we have an accurate, or at least repeatable, method of defining an outline, we then move in to the landmark selection. I have absolutely no problem with the definitions or use of the true landmarks. I was however a little confused by the simultaneous use and non-use of semilandmarks. As I understand it, semilandmarks were obtained for all the prints to represent the outlines, then were analysed to find the mean outline shape. This mean outline shape was then simply used for visualisation, not subsequent analysis.

Why?

If you have the semilandmarks, why not analyse the shape in full? I appreciate that the some outlines may create noise or be overly influenced by abiotic concerns, and that would be a valid reasoning. What is then not valid is to interpret shape changes to the warps of the mean outline in the subsequent analysis of only true landmarks, as can appear to be the case in the results and in Figs 9 and 12 (also, where is Fig. 11…?). I believe that you know that this is wrong, and are only using the mean outlines to help visualise. But, by mixing the geometric morphometrics and ichnological descriptions together, it comes across badly; a reader who is unfamiliar with morphometrics is likely to be at best confused (as I was for a long time!), and at worst misled by the figures.

In section 3.2 (Definition of footprint margins), my main issues are not with the methods but the interpretation, and the fact that this is arguably the most important facet of the paper that largely gets overlooked! For the methods though, I wanted to ask whether one person performed all the interpretational outlines, or whether you’ve recorded the variation caused by multiple people subjectively interpreting the same outline?
The same goes for placing the landmarks, now I think about it! Were these all placed by the same individual, and if not, how important is inter-user error? (it’s probably quite small and not that important, but it’s worth checking just in case)

Minor comments:
Line 161: “optimal number of semilandmarks” – optimal for what? Representing the curve, or reducing the number of analytical points without sacrificing precision?
Line 164: Did you slide the semilandmarks as part of the GPA, and if not, why not? This is a pretty easy step in tpsDig/tpsRelw, or alternatively, the Geomorph R package.
Fig 10: A lot of people in morphometrics circles hate vector plots, and with good reason: they are messy, and imply motion from one shape to another, when actually what the analysis represents is variability. The blue mean/range images are fine, suggest replacing the green ones with mean/PC1max/PC1min outlines.
Fig 12: Why are you only showing deviation from the mean in one direction? Each outline drawing should have two lines in addition to the mean, showing the representative shape at the maximum and minimum ends of the depicted PC.

Validity of the findings

There are some worthwhile morphometrics results in this manuscript, even though they feel rather difficult to find. Quantifying the variability on each landmark (Final paragraph of 4.3; Fig. 13; largely absent from discussion) is a great way of illustrating potentially noisy areas of the print, which you rightly say should be avoided when trying to take reference measurements. Another key finding is that ornithopod and theropod tracks aren’t really distinguishable without well-preserved ungal features, clearly demonstrated in the PCA of Fig. 9, and in Fig. 14. Even the CVA doesn’t do a great job of pulling them apart, and used alone is no good as a diagnostic tool for unknown prints anyway because of the need to define group membership a priori.

Section 5.3 (Definition of footprint margins: comparison of results)
This is, to my mind, the most important topic in the entire paper: how to define footprints objectively for further comparison, in such a way that can be applied to any track at any site. Unfortunately, I can see that answering this is beyond the scope of your study, but the potential to solve this problem with morphometrics is high, and could be better worked in to the discussion.
The major problem that I have with this section is the conflation of precision with accuracy. This first appears in the results (line 497-499), “the total variance is markedly higher … suggesting that the capturing of the foot anatomy is much poorer compared to the interpretive approach”.

Just because the objective method has higher variance than the subjective method, it does not mean it is less accurate. Imagine throwing darts at a dartboard: you could hit exactly the same spot every time, but still miss the bullseye (high precision, low accuracy), or you could cluster around the bullseye but never hit the same spot (low precision, high accuracy). The degree of variance helps you to see which method is most repeatable (a useful thing to know, for sure), but it in no way dictates which is the most accurate representation of foot anatomy. The only way to know this would be to experimentally validate these methods on prints from extant animals with known foot soft-tissue morphology, probably in a range of substrate conditions. This is why the present study cannot fully address the problem, but if no-one’s doing this yet, there’s a grant in there…

That said, provided that a repeatable outline definition method has been justified and is employed consistently within a given study, it will be fine to proceed to analyse a trackway using morphometrics. As I mentioned above, I would be curious to know how variable the subjective approach truly is if the outlines were determined by different people.

Additional comments

I think that there’s a lot of good material here but the length of the manuscript, coupled with the tendency to mix quantitative morphometrics with qualitative descriptions, makes it incredibly difficult to read and appreciate. Please consider breaking this in to two papers! Methodologically, the work is largely sound but does need some more information, and to be presented in a much clearer way to ensure that the morphometrics are not misleading or misinterpreted. Finally, have another think about where the weight of the discussion needs to lie. This would be much easier if the manuscript was divided in two.

·

Basic reporting

No Comments

Experimental design

No Comments

Validity of the findings

No Comments

Additional comments

Dear Authors,
the manuscript is well written and the sample number, and the analyses made are more than sufficient to validate your conclusions. I am very glad that some developments are being made in the application of GM to dinosaur ichnology. For these reasons I suggest the publication of the manuscript after minor revisions.

My main concern about the method is the application of the 30 contour-lines method. It is not clearly stated why this should be better than a known-pace contours, which in my opinion is applicable also to tracks, form different locations, with a different depth. 12th and 11th lines may be good for that specific trackway of Münchehagen, but can be applied to other specimens. Thus I suggest to better support the method chosen and/or make a comparison with other contour-line possibilities.

I suggest to upload on Figshare also the photos use to generate the 3D models. This will allow not only future comparisons, but it could be important to generate better 3D models accordingly with the software improvement (in my personal experience, I can make much better and more accurate models now than 2 years ago, starting from the same images)

Other comments, suggestions and corrections are in the attached pdf.

Best regards
Matteo Belvedere

---

## Round 0.2 · accepted · Accept

Thank you for taking the reviews seriously and for working to make the paper more accessible and ultimately publishable.